# Enhanced Effects of Iron on Mycelial Growth, Metabolism and In Vitro Antioxidant Activity of Polysaccharides from *Lentinula edodes*

**DOI:** 10.3390/bioengineering9100581

**Published:** 2022-10-19

**Authors:** Quanju Xiang, Huijuan Zhang, Xiaoqian Chen, Shiyao Hou, Yunfu Gu, Xiumei Yu, Ke Zhao, Xiaoping Zhang, Menggen Ma, Qiang Chen, Penttinen Petri, Xiaoqiong Chen

**Affiliations:** 1College of Resource, Sichuan Agricultural University, Chengdu 611130, China; 2Rice Research Institute, Sichuan Agricultural University, Chengdu 611130, China

**Keywords:** *Lentinula edodes*, iron, polysaccharides, enzymatic activity, gene expression, antioxidant activity

## Abstract

The polysaccharides found in *Lentinula edodes* have a variety of medicinal properties, such as anti-tumor and anti-viral effects, but their content in *L**. edodes* sporophores is very low. In this study, Fe^2+^ was added to the liquid fermentation medium of *L**. edodes* to analyze its effects on mycelial growth, polysaccharide and enzyme production, gene expression, and the activities of enzymes involved in polysaccharide biosynthesis, and in vitro antioxidation of polysaccharides. The results showed that when 200 mg/L of Fe^2+^ was added, with 7 days of shaking at 150 rpm and 3 days of static culture, the biomass reached its highest value (0.28 mg/50 mL) 50 days after the addition of Fe^2+^. Besides, Fe^2+^ addition also enhanced intracellular polysaccharide (IPS) and exopolysaccharide (EPS) productions, the levels of which were 2.98- and 1.79-fold higher than the control. The activities of the enzymes involved in polysaccharides biosynthesis, including phosphoglucomutase (PGM), phosphoglucose isomerase (PGI), and UDPG-pyrophosphorylase (UGP) were also increased under Fe^2+^ addition. Maximum PGI activity reached 1525.20 U/mg 30 days after Fe^2+^ addition, whereas PGM and UGP activities reached 3607.05 U/mg and 3823.27 U/mg 60 days after Fe^2+^ addition, respectively. The Pearson correlation coefficient showed a strong correlation (*p* < 0.01) between IPS production and PGM and UGP activities. The corresponding coding genes of the three enzymes were also upregulated. When evaluating the in vitro antioxidant activities of polysaccharides, EPS from all Fe^2+^-treated cultures exhibited significantly better capacity (*p* < 0.05) for scavenging -OH radicals. The results of the two-way ANOVA indicated that the abilities of polysaccharides to scavenge O_2_^−^ radicals were significantly (*p* < 0.01) affected by Fe^2+^ concentration and incubation time. These results indicated that the addition of iron provided a good way to achieve desirable biomass, polysaccharide production, and the in vitro antioxidation of polysaccharides from *L. edodes*.

## 1. Introduction

*Lentinula edodes*, commonly known as the shiitake mushroom, has great nutritional and medical value. It contains the bio-active polysaccharide lentinan, a β-(1,6)-(1,3)-glucan that offers multiple biological benefits, including anti-tumor, anti-oxidation, and immunity-enhancing activities [1,2,3,4,5]. Lentinan is normally extracted from the fruiting body of *L. edodes*. However, the cultivation of *L. edodes* takes a relatively long time, and there are great uncertainties with regard to yield and quality. The cultivation model of *L. edodes* has changed from logs to bags, shortening the cultivation time to some extent and increasing the yield of fruiting bodies [6]. In addition, extracting and purifying polysaccharides from liquid fermentation shows the potential for shortening the production cycle.

However, the composition and content of polysaccharides in *L. edodes* depend on the growth conditions. The polysaccharide content of fungi varies with carbon and nitrogen sources, temperature, pH, developmental stages, and other cultivation measures [7,8]. In addition, exogenous substances, such as chemicals or ions, affect secondary metabolic pathways. The addition of Mg^2+^ and Mn^2+^ has been reported to enhance the mycelial growth and extracellular polysaccharide (EPS) content of *Cordyceps militaris* [9], whereas Na^+^ and Ca^2+^ have been shown to promote the intracellular polysaccharide (IPS) and EPS content of *L. edodes* in liquid fermentation [10]. The addition of Ca^2+^ ions, coupled with blue light irradiation, promotes EPS production in *Agaricus sinodeliciosus* [11]. Thus, it is feasible to promote polysaccharide production by adding different exogenous substances.

Exogenous substances can affect the biosynthesis of polysaccharides, but whether these substances will also affect the biological activities of the polysaccharides remains unclear. The biological activities of polysaccharides depend on many factors, including water solubility, molecular weight, chemical conformation, and the appropriately substituted ionized groups [12,13,14]. Recently, many reports have focused on the selenylation of polysaccharides from *L. edodes* for achieving more efficient bioactivities [14,15,16]. Se-enriched polysaccharides, produced by cultivating *L. edodes* in selenium-containing medium, inhibited the proliferation of human peripheral blood mononuclear cells, down-regulated the production of tumor necrosis factor, and reduced the cytotoxic activity of human natural killer cells [17]. Se-enriched polysaccharides from *Cordyceps gunnii* affected the viability of SKOV-3 cells, H1299 cells, and HepG2 cells and inhibited ovarian tumors in a rat model [18]. These findings suggest that the addition of exogenous substances affects the activities of polysaccharides. However, whether other exogenous ions apart from selenium have any effect on the activity of polysaccharides has yet to be reported.

Iron is an essential trace element in the human body and is an important component of hemoglobin. As with other ions, iron promoted the mycelial growth of fungi [19], and iron-polysaccharide complexes had high antioxidant activity, equal to that of vitamin C [20,21]. To the best of our knowledge, this is the first study evaluating the effects of iron on polysaccharide biosynthesis and biological activities in *L. edodes*. We aimed at exploring the effects of iron on *L. edodes* with regard to growth, polysaccharide production, enzyme activities, gene expression of the enzymes involved in polysaccharide biosynthesis, and the in vitro antioxidant activities of polysaccharides. Our findings will provide a theoretical basis to promote polysaccharide production in *L. edodes*.

## 2. Materials and Methods

### 2.1. Fungal Strain and Culture Conditions

The strain *L. edodes* 808 (deposited in the Agricultural Culture Collection of China with the number 52357), from the Chengdu Academy of Agriculture and Forestry Sciences, was cultured in sterilized potato dextrose agar (PDA) medium for 10 days at 25 °C until the mycelium covered the whole surface of the medium, and mycelial plugs (diameter: 5 mm) were used as inoculum. Three mycelial plugs were inoculated into 250 mL Erlenmeyer flasks containing 50 mL synthetic medium (35 g glucose, 5 g peptone, 2.5 g yeast extract, 1 g KH_2_PO_4_·H_2_O, 0.5 g MgSO_4_·7H_2_O, 0.05 g vitamin B_1_, and distilled water to make the final volume of 1 L, then sterilized at 121 °C for 30 min) as described by Adil et al. [10].

### 2.2. Assessment of Optimal Fe^2+^ Concentration and Induction Time

To determine the optimal induction concentration of Fe^2+^, five different concentrations (0, 50, 200, 600, and 1000 mg/L) of Fe^2+^ (as FeCl_2_) were added to the synthetic medium at inoculation. Samples were collected after 50 days of static culture at 25 °C. Subsequently, 200 mg/L of Fe^2+^ (as FeCl_2_), which was determined to be the optimal induction concentration, was added to the synthetic medium at four time points: at inoculation (I), after 3 days of static culture (II), after 7 days of shaking at 150 rpm (III), or after 7 days of shaking at 150 rpm and 3 days of static culture (IV). After Fe^2+^ addition, the cultures were incubated under static conditions at 25 °C until the culture time reached 50 days. Experiments were performed in triplicate. As a control, cultures were performed in the liquid medium without Fe^2+^, incubated under static conditions at 25 °C for 50 days. Samples were collected for the determination of biomass and polysaccharide content.

### 2.3. Biomass and Polysaccharides Determination

Fungal biomass was collected by filtration through a 100-mesh screen. The collected mycelium was washed with distilled water (50 mL) three times, trapped on a pre-dried and pre-weighed filter paper, and dried at 60 °C until a constant weight was achieved.

For IPS extraction, 0.2 g of dried mycelium was ground into a powder with liquid nitrogen, then 10 mL of boiling water was added. In order to prevent water evaporation and optimize IPS extraction, a reflux device was used. After 1 h, the aqueous phase was collected by centrifugation (Eppendorf centrifuge 5804R, Eppendorf, Hamburg, Germany, 3000 rpm, 30 min, 4 °C). The residual pellet was extracted twice. Four volumes of cold 95% ethanol were added to the supernatant and kept overnight at 4 °C. After centrifugation (3000 rpm, 30 min, 4 °C), the supernatant was discarded, the excess ethanol was evaporated, and the precipitate IPS was dissolved in 2 mL water.

For EPS extraction, 30 mL of cold 95% ethanol was added to 10 mL of culture filtrate and incubated at 4 °C for 12 h. EPS was then extracted by centrifugation at 3000 rpm for 30 min at 4 °C. After removing and evaporating the supernatant, the precipitate was dissolved in 2 mL water.

The polysaccharide content was determined following the phenol-sulfuric acid method, with glucose as the standard [22]. The IPS content was then expressed as a percentage of biomass (wt %), and EPS was expressed in milligrams of polysaccharides per milliliter (mg/mL). The total polysaccharide (TPS) content was calculated using the following formula: TPS (mg) = IPS (wt %) × biomass (mg) + EPS (mg/mL) × 50 mL.

### 2.4. Dynamic Changes in Biomass and Polysaccharides Production

For evaluating further dynamic changes, Fe^2+^ was added to the synthetic medium at the optimal induction concentration (200 mg/L) for the optimal time (7 days of shaking at 150 rpm and 3 days of static culture) and incubated at 25 °C for a further 60 days. Samples were collected every 15 days and analyzed for mycelial biomass, IPS, EPS, and TPS contents.

### 2.5. Effects of Fe ^2+^ on Polysaccharide Synthesis

To investigate the effects of Fe^2+^ on the activities of enzymes involved in *L. edodes* polysaccharide biosynthesis and to determine the transcriptional expression levels of their encoding genes, cultures were performed with the optimal induction concentration for the appropriate induction time. After the addition of Fe^2+^, cultures were incubated under static conditions at 25 °C for a further 60 days. Samples were collected every 15 days.

#### 2.5.1. Enzyme Activity Assays

Fresh mycelium (0.1 g), collected as described in Section 2.3, was washed three times with 1 mL of 20 mM phosphate buffer (pH 6.5), powdered in liquid nitrogen, and dissolved in 1 mL phosphate buffer (20 mM, pH 6.5). After centrifugation (10,000 rpm, 15 min, 4 °C), the supernatant was used as the crude enzyme solution. The activities of phosphoglucose isomerase (PGI), phosphoglucomutase (PGM), and UDPG-pyrophosphorylase (UGP) were determined, as previously described [10].

#### 2.5.2. Transcriptional Expression Analysis

The total RNA was isolated from fresh mycelium using Trizol reagent (Sangon Biotech, Shanghai, China) according to the manufacturer’s instructions. The purity and quantity of the extracted RNA were assessed using a nano spectrophotometer (Nanodrop-1000, Thermo Scientific, Waltham, MA, USA), and the integrity was evaluated via agarose gel electrophoresis. cDNA was synthesized from 1.5 μg of total RNA using a reverse transcription kit (Tiangen, Beijing, China) according to the manufacturer’s instructions.

Before performing a quantitative real time polymerase chain reaction (qRT-PCR) assay, the cDNA was diluted 10 times with nuclease-free water. The expression levels of *pgi*, *pgm*, and *ugp* were quantified using the same primers as previously described [10]. The qRT-PCR assays were performed in 96-well plates with SYBR Green detection in an iCycler iQ5 thermocycler (Bio-Rad, Hercules, CA, USA). The reaction included 1 μL cDNA, 0.5 μL of each primer (1 μM), 10 μL Cham Q Universal SYBR qPCR Master Mix (Vazyme, Nanjing, China), and nuclease-free water to make the final volume of 20 μL. Nuclease-free water served as a negative control. Reactions were performed under the following conditions: 95 °C for 5 min, followed by 40 cycles of 95 °C for 15 s, 60 °C for 15 s, and 72 °C for 30 s. The melting curves were set to ensure amplification product specificity. The qRT-PCR products were confirmed by agarose gel electrophoresis and gene sequencing (Sangong Biotech, Shanghai, China). The expression levels of genes were normalized using *Rpl4* as an internal control [23]. The relative expression level was calculated using the formula Y = 10△Ct/3 × 100% [24], where △Ct is the difference in the cycle threshold value of the target gene (*pgi*, *ugp**,* and *pgm*) and *Rpl*4. Mean values were obtained from three biological replicates.

### 2.6. In Vitro Antioxidant Activities of Polysaccharides

To analyze whether the polysaccharides extracted from the iron-containing medium or mediums incubated for different incubation times have different biological activities, the in vitro antioxidant activities of IPS and EPS were analyzed. Cultures were performed, as reported in Section 2.2 and Section 2.4.

In vitro antioxidant activities were evaluated according to their ability to scavenge the superoxide anion (O_2_^−^), hydroxyl (-OH), and 1,1-diphenyl-2-picrylhydrazyl (DPPH) radicals. The superoxide anion and O_2_^−^ radical scavenging abilities were determined following the o-phenanthroline and pyrogallol autoxidation methods. The DPPH scavenging activity was measured as described previously [25]. Vitamin C, at concentrations of 0.12, 0.24, 0.36, 0.48, and 0.60 mg/mL, was assayed as a positive control. Deionized water instead of polysaccharide solution was used as the negative control.

To determine the -OH scavenging ability, 200 μL polysaccharide solution was mixed with 60 μL FeSO_4_ (6 mmol/L), salicylic acid ethanol solution (6 mmol/L), and 60 μL H_2_O_2_ (0.1%). The mixture was incubated in a 37 °C water bath for 30 min, after which the absorbance was measured at 510 nm. The scavenging rate was calculated using the following formula: scavenging rate (%) = [1 − (A_1_ − A_2_)/A_3_] × 100%, where A_1_ is the absorbance of the polysaccharide-containing solution, A_2_ is the absorbance measured with anhydrous ethanol instead of salicylic acid, and A_3_ is the absorbance of the negative control.

To determine the O_2_^−^ scavenging ability, a mixture of 900 μL phosphate buffer (0.01mol/L), 200 μL polysaccharide, and 80 μL pyrogallol solution (25 mmol/L) was incubated in a 25 °C water bath for 5 min. Following this, 200 μL of hydrochloric acid (80 mmol/L) was added to stop the reaction. The absorbance was measured at 420 nm after placing the mixture for 3 min in the dark. The scavenging rate was calculated using the following formula: scavenging rate (%) = [1 − (A_1_ − A_2_)/A_3_] × 100%, where A_1_ is the absorbance of the polysaccharide-containing solution, A_2_ is the absorbance measured using the corresponding solvent instead of pyrogallol, and A_3_ is the absorbance of the negative control.

To determine the DPPH scavenging ability, a mixture of 150 μL polysaccharide and 200 μL DPPH ethanol solution (0.04 mM/L) was shaken well and kept in the dark for 30 min. The absorbance was measured at 515 nm. The scavenging rate was calculated using the following formula: scavenging rate (%) = [1 − (A_1_ − A_2_)/A_3_] × 100%, where A_1_ is the absorbance of the polysaccharide-containing solution, A_2_ is the measured absorbance of the mixture, using absolute ethanol instead of DPPH solution, and A_3_ is the absorbance of the negative control.

### 2.7. Statistical Analysis

Data are expressed as means ± standard deviation (SD), obtained from triplicate experiments. Data analysis was performed using Statistical Product and Service Solutions (SPSS, version 17.0, Chicago, IL, USA). Means were compared using the least significant difference (LSD) test at *p* ≤ 0.05. Pearson’s correlation coefficient (r^2^, ranging from +1 to −1) was performed using SPSS software to demonstrate the correlation between polysaccharide content and enzyme activities. The two variables were considered uncorrelated (r = 0), positively (r > 0), or negatively (r < 0) correlated.

## 3. Results

### 3.1. Assessment of Optimal Fe^2+^ Concentration and Induction Time

The biomass, IPS, EPS, and TPS contents of *L. edodes* mycelium varied with Fe^2+^ concentrations (Figure 1A–C). From 0 to 200 mg/L, the mycelium biomass content increased with increasing Fe^2+^ concentration and reached the highest value at 200 mg/L, which was approximately 1.8 times higher than that of the control. However, when the concentrations ranged from 600 to 1000 mg/L, the biomass content was found to be lower than that of the control (Figure 1A). The IPS content showed a similar trend to the biomass (Figure 1B). Maximum EPS and TPS contents were also observed following treatment with 200 mg/L Fe^2+^, which were approximately 1.29 and 1.45 times higher than those of the control, respectively (Figure 1B,C). Based on the findings on biomass and polysaccharide production, the optimal induction concentration of Fe^2+^ was 200 mg/L.

The experiment was performed to determine the optimal induction time and showed (Figure 1D–F) that the highest biomass content was observed when Fe^2+^ was added after static culture for 3 days (Figure 1D). The highest IPS and EPS contents were observed when Fe^2+^ was added at time points III (after 7 days of shaking at 150 rpm) and IV (after 7 days of shaking at 150 rpm and 3 days of static culture), respectively (Figure 1E). The TPS content reached its highest level when Fe^2+^ was added at time point IV (Figure 1F).

### 3.2. Dynamic Changes in the Mycelial Biomass and Polysaccharides Production

Data on dynamic changes in mycelial growth and polysaccharide production under optimal Fe^2+^ concentration (200 mg/L) and induction time (7 days of shaking at 150 rpm and 3 days of static culture) are reported in Figure 2. The biomass and IPS contents of *L. edodes* were higher with Fe^2+^ than in the control throughout the incubation time and were 1.53 and 2.98 times higher with Fe^2+^ than those in the control, respectively (Figure 2). The highest EPS and TPS contents were observed 45 days after Fe^2+^ addition, which were 1.79 and 2.40 times higher than those in the control, respectively (Figure 2).

### 3.3. Effects of Fe^2+^ on Polysaccharide Synthesis

#### 3.3.1. Enzyme Activity Assays

PGI activity remained relatively stable at approximately 400 U/mg without Fe^2+^, whereas, following Fe^2+^ treatment, PGI activity reached its maximum value (1525.20 U/mg) 30 days after Fe^2+^ addition, and decreased thereafter (Figure 3A).

PGM activity showed only a slight increase in both the control and Fe^2+^ treated cultures, from 15 to 45 days after Fe^2+^ addition; however, the activity level increased to 3607.05 U/mg at 60 days after Fe^2+^ treatment, which was approximately twice higher than that in the control (Figure 3B).

UGP activity increased over culture time with Fe^2+^ treatment, reaching the maximum value of 3823.27 U/mg 60 days after Fe^2+^ addition, which was 4.17 times higher than that of the control. The Pearson correlation coefficient analysis shows a significant positive (r^2^ > 0.85, *p* < 0.01) correlation between IPS content and PGM and UGP activities (Table 1).

#### 3.3.2. Transcriptional Expression Analysis

The expression levels of *pgm* were higher than those of *pgi* and *ugp* (Figure 4). The relative expression of *pgi* gradually increased to the maximum level, 45 days after Fe^2+^ addition, which was 3.33 times higher with Fe^2+^ than in the control (Figure 4A).

The expression profiles of *pgm* with Fe^2+^ were similar to those of the control, while the highest expression was observed 60 days after Fe^2+^ addition (Figure 4B).

The expression levels of *ugp* were downregulated with the extension of culture time, with Fe^2+^ addition; 15 days after Fe^2+^ addition, *ugp* expression levels were 4.19 times higher than in the control (Figure 4C).

### 3.4. Effect of Fe^2+^ on the In Vitro Antioxidant Activity of Polysaccharides

The in vitro antioxidant activities of polysaccharides extracted from different concentrations of Fe^2+^ treatments are presented in Figure 5. IPS extracted from the culture filtrates obtained with 50 mg/L Fe^2+^ showed moderate antioxidant activity with regard to scavenging effects on DPPH radicals at polysaccharide concentrations of 0.36–0.60 mg/mL, compared with that extracted from the control cultures without Fe^2+^ addition. At a concentration of 0.36 mg/ml, 60% and 15% of DPPH radicals were inhibited by IPS extracted from 50 mg/L Fe^2+^ treatment and the control, respectively (Figure 5A). In tests evaluating the scavenging abilities regarding superoxide anion (O_2_^−^) radicals, IPS extracted from the medium with 200 mg/L Fe^2+^ show statistically significant antioxidant activity (Figure 5B). EPS from all Fe^2+^-treated samples exhibited significantly better scavenging effects (*p* < 0.01) on -OH radicals (Figure 5C). The two-way ANOVA showed that the effect of Fe^2+^ on the O_2_^−^ radical scavenging ability of IPS is significant (*p* < 0.01).

The in vitro antioxidant activities of polysaccharides produced in the *L. edodes* liquid cultures under optimal Fe^2+^ concentration and induction time are presented in Figure 6. IPS extracted at 30 and 45 days showed higher –OH and O_2_^−^ radical scavenging abilities, respectively, compared with the sample collected at 15 days (Figure 6A,B). Overall, IPS showed poor ability in terms of scavenging DPPH radicals (Figure 6C). EPS from cultures of all the incubation times showed a strong ability to scavenge −OH radical (Figure 6D), even at a very low concentration (0.12 mg/L). However, poor DPPH and O_2_^−^ radical scavenging abilities were observed in all the EPS (Figure 6E,F). Two-way ANOVA showed that incubation time had a significant effect on the O_2_^−^ radical scavenging effect of polysaccharides (*p* < 0.01). 

## 4. Discussion

*L. edodes*, a form of precious medicinal and edible mushroom that is rich in bioactive constituents, particularly polysaccharides, is one of the most studied mushrooms. Polysaccharides from *L. edodes* have many activities, such as anti-tumor and anti-oxidation activity, and improve human immunity [26]. However, the low content and high price limit its application. Various methods, such as strain screening [27], culture condition modifications [28], and cultivation management [29] have been applied to enhance polysaccharide production.

Exogenous ions affect the secondary metabolism of organisms by participating in various enzymatic reactions and can promote the biosynthesis of polysaccharides. The addition of Mn^2+^, Ca^2+^ and Na^+^ has been shown to result in a 2.2-, 3.7-, and 2.8-fold improvement in total ganoderic acids production [30,31,32]. Furthermore, adding 2 ‰ Fe^2+^ to the seed liquid medium can significantly increase the polysaccharide yield of *Ganoderma lucidum* [33]. Iron is necessary for microorganisms to maintain normal growth and development. As is consistent with this finding, with the addition of 200 mg/L of Fe^2+^, the mycelium grew stronger, with higher biomass and polysaccharide production in *L. edodes*. Numerous researchers have found that the same species of fungus, depending on the strain and growing conditions (e.g., the composition of the medium), may secrete exopolysaccharides of different structures or in different amounts into the medium [34,35]. The polysaccharide production of *Tricholoma mongolicus* was increased by adding metal ions in the middle and later stages of fermentation [36]. Accordingly, in this study, the maximum total polysaccharides were obtained when Fe^2+^ was added after 7 days of shaking at 150 rpm, followed by 3 days of static culture.

The lack of industrial polysaccharide production was the main drawback limiting the biological application of *L. edodes*; therefore, it was essential to improve the yield of polysaccharides from *L. edodes*. The biosynthesis of polysaccharides involves a complex regulatory network in which some enzymes play key roles. PGM, PGI, and UGP are considered important enzymes in the polysaccharide biosynthesis pathway [37,38]. Increasing UGP activity has been reported to be beneficial to pullulan production with *Aureobasidium pullulans* [37]. Our study demonstrated that the levels of these key enzymes involved in polysaccharide synthesis, as well as their gene expressions, were all enhanced after Fe^2+^ addition, while a strong positive correlation was found between the activities of PGM and UGP and IPS production. This is similar to the report that the levels of activities of PGM, PGI, and UGP were highly correlated with the amount of IPS production in *Cordyceps militaris* [9]. It may be helpful for further research on the pathway of polysaccharide biosynthesis, aimed at improving the IPS production of *L. edodes*.

Recently, polysaccharides complexed with metal ions have been reported to possibly be used to improve bioactivities [39]. At present, reports on the chemical-containing polysaccharides mainly focus on selenium-enriched polysaccharides. Se-enriched polysaccharides extracted from the *L. edodes* mycelial culture following Se treatment show enhanced antioxidant activity. Further studies have indicated that the higher activity level was because of the high amounts of organic Se compounds embedded in the polysaccharide [40]. Furthermore, Se-enriched *L. edodes* EPS showed significantly enhanced cell viability and protected the cells from oxidative stress conditions [15], implying that selenium-containing polysaccharides show higher biological activities. Iron is one of the essential trace elements in the human body and forms a core part of hemoglobin and myoglobin, which are essential for the normal function of an organism. In vitro digestive system model experiments have shown that the solubility, stability, and absorption capacity of *Flammulina velutipes* iron-containing polysaccharides are better than those of FeSO_4_ [21]. The iron-containing polysaccharide complex from *Auricularia auricular* also has higher antioxidant activity [20]. The results in this study demonstrated that EPS extracted from an iron-treated culture exhibited a significantly better (*p* < 0.05) ability of scavenging -OH radicals, suggesting that polysaccharides with or without Fe^2+^ treatment may be structurally different; this finding is worthy of further study.

## 5. Conclusions

The addition of Fe^2+^ in submerged liquid fermentation increased the biomass and polysaccharide contents of *L. edodes*. The activities of the enzymes (PGM, PGI, and UGP) involved in polysaccharide biosynthesis and the transcriptional expression profiles of their encoding genes were enhanced following Fe^2+^ treatment. Furthermore, EPS from Fe^2+^ treated samples exhibited a significantly better (*p* < 0.05) ability in terms of the scavenging ability on -OH radicals, while the concentrations of Fe^2+^ significantly (*p* < 0.01) affected the O_2_^−^ radical scavenging of polysaccharides. In conclusion, the addition of iron was a good means by which to achieve desirable biomass, polysaccharide biosynthesis, and biological activity in *L. edodes*.

## Figures and Tables

**Figure 1 bioengineering-09-00581-f001:**
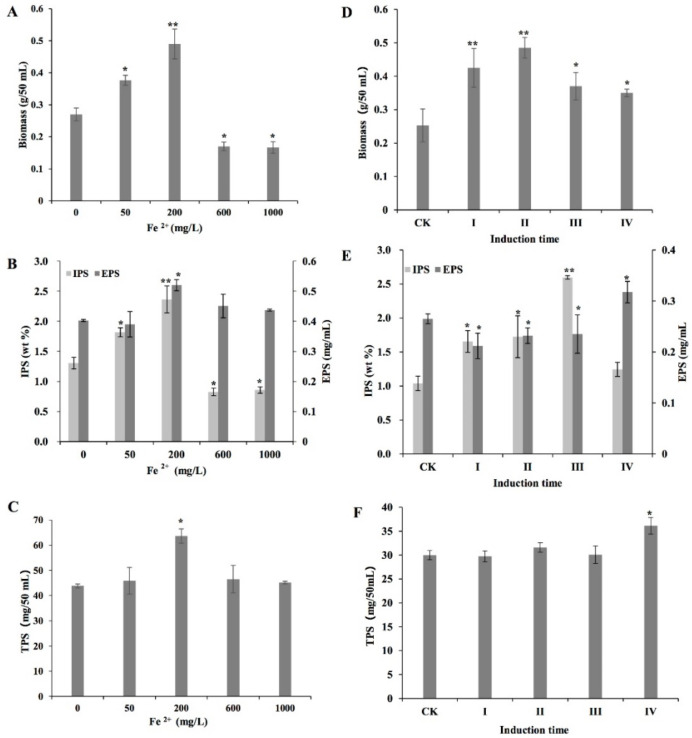
Effects of different Fe^2+^ concentrations (**A**–**C**) and induction times (**D**–**F**) on biomass, extracellular polysaccharide (EPS), intracellular polysaccharide (IPS) and total polysaccharide (TPS) production by *Lentinula edodes* strain 808 (ACCC 52357). CK: control, cultures in a liquid medium without Fe^2+^. Fe^2+^ was added as FeCl_2_: (I) at the time of inoculation; (II) after 3 days of static culture; (III) after 7 days of shaking at 150 rpm; and (IV) after 7 days of shaking at 150 rpm and 3 days of static culture. After the addition of Fe^2+^, the cultures were grown in static conditions. All cultures were incubated at 25 °C. Data were recorded after 50 days. Values are the means of experiments in triplicate ± SD. Asterisks indicate a significant statistical difference at *p* < 0.01 (**) and *p* < 0.05 (*), according to the least significant difference (LSD) test.

**Figure 2 bioengineering-09-00581-f002:**
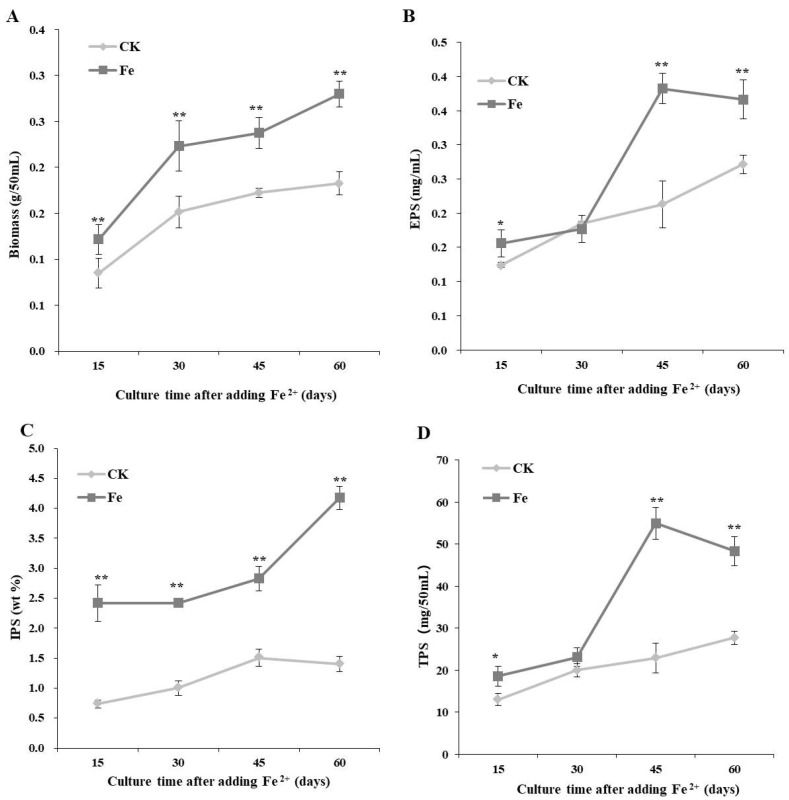
Dynamic changes in biomass (**A**), extracellular polysaccharide (EPS, (**B**)), intracellular polysaccharide (IPS, (**C**)), and total polysaccharide (TPS, (**D**)) production by the *Lentinula edodes* strain 808 (ACCC 52357) over 70 days of static cultures. CK: control without Fe^2+^; Fe: as FeCl_2_, 200 mg/L of Fe^2+^ was added after 7 days of shaking at 150 rpm and 3 days of static culture. The time on the abscissa represents the incubation time after the addition of Fe^2+^. Data are the means of experiments in triplicate. Asterisks indicate a significant statistical difference at *p* < 0.01 (**) and *p* < 0.05 (*), according to the LSD test.

**Figure 3 bioengineering-09-00581-f003:**
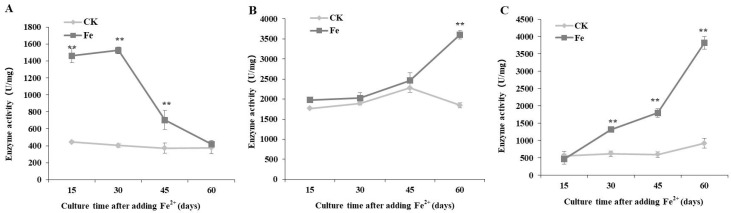
Dynamic changes in phosphoglucose isomerase (**A**), phosphoglucomutase (**B**), and UDPG-pyrophosphorylase (**C**) production by *Lentinula edodes* 808 (ACCC 52357) over 60 days of static cultures after Fe^2+^ addition. CK: control without Fe^2+^ addition; Fe: as FeCl_2_, 200 mg/L of Fe^2+^ added after 7 days of shaking at 150 rpm and 3 days of static culture. Data are the means of experiments in triplicate ± SD. Asterisks indicate a significant statistical difference at *p* < 0.01 (**), according to the LSD test.

**Figure 4 bioengineering-09-00581-f004:**
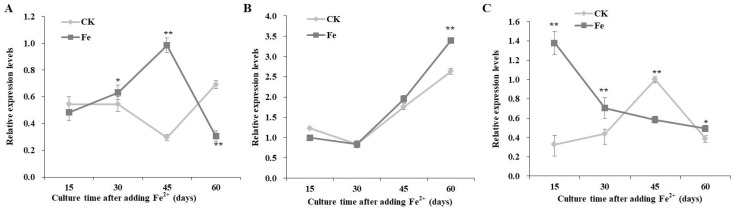
Transcriptional expression levels of *pgi* (**A**), *pgm* (**B**), and *ugp* (**C**) in *Lentinula edodes* 808 (ACCC 52357) during 60 days of static cultures after Fe^2+^ addition. CK: control without Fe^2+^ addition; Fe: as FeCl_2_, 200 mg/L of Fe^2+^ added after 7 days of shaking at 150 rpm and 3 days of static culture. Data are the means of experiments in triplicate ± SD. Asterisks indicate a significant statistical difference at *p* < 0.01 (**) and *p* < 0.05 (*), according to the LSD test.

**Figure 5 bioengineering-09-00581-f005:**
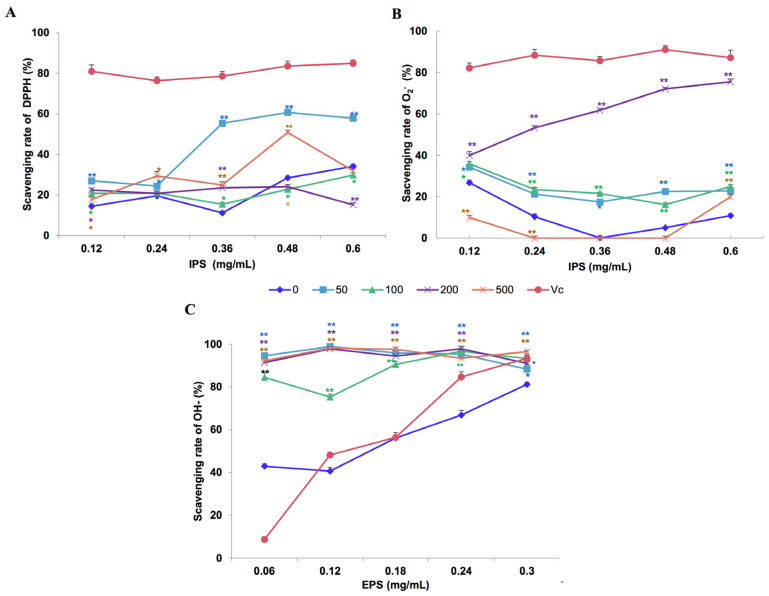
In vitro scavenging effects of intracellular polysaccharide (IPS; (**A**,**B**)) and extracellular polysaccharide (EPS; (**C**)) from *Lentinula edodes* 808 (ACCC 52357) for DPPH (**A**), O_2_^−^ (**B**), and –OH (**C**) radicals. Polysaccharides were extracted from liquid static cultures with different Fe^2+^ concentrations (0, 50, 100, 200, and 500 mg/L) added at the appropriate inoculation time and incubated under static conditions at 25 °C for 50 days. Data are the means of experiments in triplicate ± SD. Asterisks indicate significant differences at *p* < 0.01 (**) and *p* < 0.05 (*), according to the LSD test. Vitamin C (Vc) was assayed as a positive control. Deionized water (0) was used as a negative control.

**Figure 6 bioengineering-09-00581-f006:**
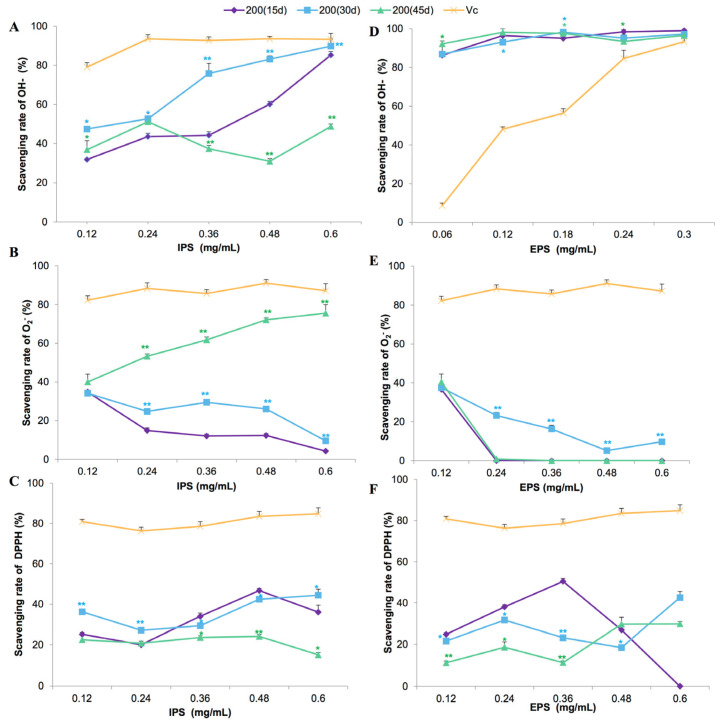
The in vitro scavenging effects of intracellular (IPS; (**A**–**C**)) and extracellular (EPS; (**D**–**F**)) polysaccharides from *Lentinula edodes* 808 (ACCC 52357) cultures on –OH (**A**,**D**), O_2_^−^ (**B**,**E**), and DPPH (**C**,**F**) radicals. Polysaccharides were extracted from liquid static cultures with 200 mg/L Fe^2+^, added after 7 days of shaking at 150 rpm and 3 days of static culture, and incubated under static conditions at 25 °C for 15, 30, and 45 days. Data are the means of three replicates ± SD. Asterisks indicate a significant difference at *p* < 0.01 (**) and *p* < 0.05 (*), according to LSD test. Vitamin C (Vc) was assayed as a positive control. Deionized water (0) was used as a negative control.

**Table 1 bioengineering-09-00581-t001:** Correlation coefficients between the intracellular polysaccharide (IPS) and exopolysaccharide (EPS) contents and phosphoglucose isomerase (PGI), phosphoglucomutase (PGM), and UDPG-pyrophosphorylase (UGP) activities.

	PGI	PGM	UGP
IPS	−0.262	0.867 **	0.869 **
EPS	−0.385	0.177	0.379

** Significant correlation at *p* < 0.01.

## Data Availability

All data are presented in the article.

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
