# Peer review of "Enhanced Effects of Iron on Mycelial Growth, Metabolism and In Vitro Antioxidant Activity of Polysaccharides from Lentinula edodes"

_bioengineering, 2022, doi:10.3390/bioengineering9100581_

Round 1
Reviewer 1 Report (Previous Reviewer 1)
Thank you for your response to my comments.
The revised version of the manuscript bioengineering-1836916-resubmitted was improved but is still very confusing in some parts.
Many of the answers given in the “Point-by-point responses to the Reviewers' comments” must be included in the manuscript.
Pay attention to the pagination. Figures 1 and 2 have the legend on two pages.
The current name of the fungus is Lentinula edodes. See Index fungorum at http://www.indexfungorum.org/names/Names.asp
Enhance abstract content.
Line 11: use “properties” instead of “effects”
Line 12: insert “sporophores” between “edodes” and “is very”
Line 13: If “metabolism” refers to the production of polysaccharides and enzymes, use “polysaccharides and enzymes production”
Line 14: use “gene expressions and activities of enzymes involved in polysaccharide biosynthesis”
Line 14: What does “bioactive ” mean in this sentence? I think “in vitro antioxidant”
Line 35: use “vulgarly” instead of “also”
Line 64: use “ovarian tumour in the model rat” instead of “tumors”
Line 68: use “component” instead of “part”
Line 80: explain the acronyms ACCC
Lines 86-87: use “Adil et al.” instead of “our previous report”
Section 2.2: Experiments performed to determine the optimal induction time are not clear.
Line 91: separate "and” and “1000”
Line 93: insert “(as FeCl2)” between “Fe2+” and “,”
Line 96: use “After Fe2+ addition, the” instead of “The”
Line 97: add “further” between “for” and “50”
Describe the control performed for induction time experiments.
Line 98: insert “Cultures in the liquid medium without Fe2+ were performed as a control.” before “Samples”
Lines 103-110: The IPS extraction procedure is not clear. Rewrite.
Line 104 and 106: What does “reflux” mean in this sentence? I appreciated your response and well know the applied procedure.
I suggest: ”For IPS extraction, 0.2 g of dried mycelium were ground into a powder with liquid nitrogen and 10 mL of boiling water were added. In order to prevent water evaporation, and optimise IPS extraction, a reflux device was used. After 1 h, the aqueous phase was collected by centrifugation (Eppendorf centrifuge 5804R, Eppendorf, Hamburg, Germany, 3000 rpm, 30 min, 4 °C). The residual pellet was extracted twice. Four volumes of cold 95% ethanol were added to the supernatant and kept overnight at 4 °C. After centrifugation (3000 rpm, 30 min, 4 °C), the supernatant was discarded, excess ethanol was evaporated, and the precipitate IPS was dissolved in 2 mL of water.
Line 112: use “collected” instead of “precipitated”
Line 120: use “polysaccharides production” instead of “polysaccharide”
Line 134: use “2.3” instead of “2.5”
Line 166: use “Sections 2.2 and 2.3” instead of “Section 2.2 and 2.5”
Line 172: use “assayed” instead of “served”
Lines 200-201: delete “Significant differences were determined by analysis of variance (ANOVA, St. Louis, 200 MA, USA) including least significant difference (LSD).”
Lines 205-207: use “Means were compared using the least significant difference (LSD) test at P ≤ 0.05. Pearson's correlation coefficient (r2 ranging from +1 to −1) was performed by SPSS software to demonstrate the correlation between polysaccharide content and enzyme activities. The two variables were considered uncorrelated (r = 0), positively (r > 0) or negatively (r < 0) correlated.” instead of “Pearson's correlation coefficient 205 was used to demonstrate the correlation between polysaccharide content and enzyme activities.”
Lines 227-228: use “Asterisks” instead of “Different letters”
Lines 228-229: use “LSD test” instead of “ANOVA-LSD”
Line 236: use “polysaccharides production” instead of “polysaccharide”
Line 238: indicate optimal induction time
Line 250: use “Asterisks” instead of “The different letters”
Line 250: use “LSD test” instead of “one –way ANOVA-LSD”
Line 253: use “3.3.a.” instead of “3.3a.”
Section 3.3.a: Insert a table with Pearson correlation coefficients between polysaccharide content and enzyme activities.
Line 270: use “Asterisks” instead of “The different lowercase letters”
Line 271: use “LSD test” instead of “one –way ANOVA-LSD”
Line 272: use “3.3.b.” instead of “3.3b.”
Line 283: use “Asterisks” instead of “The different lowercase letters”
Line 287: use “LSD test” instead of “one –way ANOVA-LSD”
Line 290: use “culture filtrates obtained ” instead of “medium”
Line 293: use “cultures” instead of “medium”
Line 297: separate "Fig. ” and “5C”
Figure 5:
improve the graphical magnification.
Use black colour for axes.
Differentiate line colours of 200 and 500, in particular for section C.
Separate “DPPH” and (%)
Line 305: separate "(C)” and “radicals”
Lines 307-310: use “Asterisks indicate significant difference at P < 0.01 (**) and P < 0.05 (*) according to LSD test. Vitamin C (Vc) was assayed as a positive control. Deionized water (0) was used as a negative control.” instead of “According … activity.”
Table 1: I prefer the arrangement as a figure.
Why positive (vitamin C) and negative (deionized water) controls are not present?
Line 321: use “Table 1. Percentage of the in vitro scavenging activity of intracellular polysaccharide (IPS) and extracellular polysaccharide (EPS) produced by Lentinula edodes 808 (ACCC 52357)*” instead of “Table 1 … cultures”
Lines 322-324: use “* Liquid cultures were performed with 200 mg/L Fe2+ added after 7 days of shaking at 150 rpm and 3 days of static culture and incubated under static conditions at 25 °C for 15, 30 and 45 days. Data are the means of triplicate experiment ± SD. Values in bold are significantly different (P < 0.01) according to the LSD test. nd: not detected” instead of “Polysaccharides … detected”
Improve the discussion section.
Lines 330-332: improve
Arrange the “Reference” list following the Journal instructions for authors.
Kind regards
Author Response
Dear editor, dear Reviewer,
Thank you for your constructive and helpful comments concerning our manuscript. We have revised our manuscript in response to your comments. Please find a detailed point-by-point reply below. Colour coding: Referee comments are in black, our responses are in blue; in the revised version of the manuscript, all changes are marked using the “Track Changes” function MS Word
Sincerely,
Quanju Xiang, on behalf of all authors
-----------------------------------------------
Point-by-point responses to the Reviewers' comments:
- Thank you for your response to my comments. The revised version of the manuscript bioengineering-1836916-resubmitted was improved but is still very confusing in some parts. Many of the answers given in the “Point-by-point responses to the Reviewers' comments” must be included in the manuscript. The revised version of the manuscript bioengineering-1836916-resubmitted was improved but is still very confusing in some parts.
Response: We are extremely grateful to the reviewer for the professional and constructive comments, which not only improved the article, but also gave us a lot. Based on these comments, we have carefully revised the manuscript, we believe it is more clear.
- Pay attention to the pagination. Figures 1 and 2 have the legend on two pages.
Response: Thanks. They have been modified.
- The current name of the fungus is Lentinula See Index fungorum at http://www.indexfungorum.org/names/Names.asp
Response: Thanks for pointing out this professional comment, and we have revised it throughout the manuscript.
- Enhance abstract content.
Line 11: use “properties” instead of “effects”
Line 12: insert “sporophores” between “edodes” and “is very”
Line 13: If “metabolism” refers to the production of polysaccharides and enzymes, use “polysaccharides and enzymes production”
Line 14: use “gene expressions and activities of enzymes involved in polysaccharide biosynthesis”
Line 14: What does “bioactive” mean in this sentence? I think “in vitro antioxidant”
Response: Thanks for the valuable comments. We have revised the abstract according to these comments.
- Line 35: use “vulgarly” instead of “also”
Line 64: use “ovarian tumour in the model rat” instead of “tumors”
Line 68: use “component” instead of “part”
Response: Thanks for the valuable comments, we have revised them according to these comments.
- Line 80: explain the acronyms ACCC
Response: Thanks. The full name of ACCC has been added in the revised manuscript as “The strain L. edodes 808 (deposited in the Agricultural Culture Collection of China with the number of 52357) from…..”
- Lines 86-87: use “Adil et al.” instead of “our previous report”
Response: Thanks. It has been revised.
- Section 2.2: Experiments performed to determine the optimal induction time are not clear.
Response: Thanks. We are so sorry for the unclear statement. Microorganism at various growth stages is differentiated in their susceptibility to environmental stresses. In analyzing the most appropriate Fe2+ addition time (optimal induction time), Fe2+ is added at four different time points. The incubation time for all the four treatments is 50 days including time before Fe2+ addition. The incubation condition and setting of control have been modified in the revised manuscript.
- Line 91: separate "and” and “1000”
Line 93: insert “(as FeCl2)” between “Fe2+” and “,”
Response: Thanks. They have been modified.
- Line 96: use “After Fe2+ addition, the” instead of “The”
Line 97: add “further” between “for” and “50”
Response: Thanks for the comments. Since the incubation time for all the four treatments is 50 days including incubation time before Fe2+ addition. The “further” is not added to this sentence, and this statement has been modified as “After Fe2+ addition, the cultures were incubated under static conditions at 25 °C until the culture time reaches 50 days.”.
- Describe the control performed for induction time experiments.
Response: Thanks. The control for the optimal induction concentration and time experiments is the same. It has been modified as “Cultures in the liquid medium without Fe2+ incubated under static conditions at 25 °C were performed as a control.”
- Line 98: insert “Cultures in the liquid medium without Fe2+ were performed as a control.” before “Samples”
Response: Thanks. It has been inserted.
- Lines 103-110: The IPS extraction procedure is not clear. Rewrite.
Line 104 and 106: What does “reflux” mean in this sentence? I appreciated your response and well know the applied procedure.
I suggest: ”For IPS extraction, 0.2 g of dried mycelium were ground into a powder with liquid nitrogen and 10 mL of boiling water were added. In order to prevent water evaporation, and optimise IPS extraction, a reflux device was used. After 1 h, the aqueous phase was collected by centrifugation (Eppendorf centrifuge 5804R, Eppendorf, Hamburg, Germany, 3000 rpm, 30 min, 4 °C). The residual pellet was extracted twice. Four volumes of cold 95% ethanol were added to the supernatant and kept overnight at 4 °C. After centrifugation (3000 rpm, 30 min, 4 °C), the supernatant was discarded, excess ethanol was evaporated, and the precipitate IPS was dissolved in 2 mL of water.
Response: Thanks for the valuable comments. The polysaccharide was extracted by water and precipitated by ethanol. In order to extract as much polysaccharide as possible, a reflux device is used. This section has been revised as follows:
“For IPS extraction, 0.2 g of dried mycelium was ground into powder with liquid nitrogen and 10 mL of boiling water were added. In order to prevent water evaporation and optimize IPS extraction, a reflux device was used. After 1 h, the aqueous phase was collected by centrifugation (Eppendorf centrifuge 5804R, Eppendorf, Hamburg, Germany, 3000 rpm, 30 min, 4 °C). The residual pellet was extracted twice. Four volumes of cold 95% ethanol were added to the supernatant and kept overnight at 4 °C. After centrifugation (3000 rpm, 30 min, 4 °C), the supernatant was discarded, excess ethanol was evaporated, and the precipitate IPS was dissolved in 2 mL water.”
At the same time, the EPS extraction procedure have been similarly modified.
- Line 112: use “collected” instead of “precipitated”
Line 120: use “polysaccharides production” instead of “polysaccharide”
Line 134: use “2.3” instead of “2.5”
Line 166: use “Sections 2.2 and 2.3” instead of “Section 2.2 and 2.5”
Line 172: use “assayed” instead of “served”
Response: Thanks for the comments, all of them have been modified.
- Lines 200-201: delete “Significant differences were determined by analysis of variance (ANOVA, St. Louis, 200 MA, USA) including least significant difference (LSD).”
Lines 205-207: use “Means were compared using the least significant difference (LSD) test at P ≤ 0.05. Pearson's correlation coefficient (r2 ranging from +1 to −1) was performed by SPSS software to demonstrate the correlation between polysaccharide content and enzyme activities. The twovariables were considered uncorrelated (r = 0), positively (r > 0) or negatively (r < 0) correlated.” instead of “Pearson's correlation coefficient 205 was used to demonstrate the correlation between polysaccharide content and enzyme activities.”
Response: Thanks. This sentence has been modified as suggested.
- Lines 227-228: use “Asterisks” instead of “Different letters”
Lines 228-229: use “LSD test” instead of “ANOVA-LSD”
Line 236: use “polysaccharides production” instead of “polysaccharide”
Line 238: indicate optimal induction time
Line 250: use “Asterisks” instead of “The different letters”
Line 250: use “LSD test” instead of “one –way ANOVA-LSD”
Line 253: use “3.3.a.” instead of “3.3a.”
Response: Thanks. All of them have been modified.
- Section 3.3.a: Insert a table with Pearson correlation coefficients between polysaccharide content and enzyme activities.
Response: Thanks. A table (named Table 1) has been added as follows:
Table 1 Correlation coefficients between polysaccharides content and PGI, PGM and UGP activities
|
PGI |
PGM |
UGP |
IPS |
﹣0.262 |
0.867** |
0.869** |
EPS |
-0.385 |
0.177 |
0.379 |
* * Significant correlation at p<0.01
- Line 270: use “Asterisks” instead of “The different lowercase letters”
Line 271: use “LSD test” instead of “one –way ANOVA-LSD”
Line 272: use “3.3.b.” instead of “3.3b.”
Line 283: use “Asterisks” instead of “The different lowercase letters”
Line 287: use “LSD test” instead of “one –way ANOVA-LSD”
Line 290: use “culture filtrates obtained ” instead of “medium”
Line 293: use “cultures” instead of “medium”
Line 297: separate "Fig. ” and “5C”
Response: Thanks, all of them have been modified.
- Figure 5:
improve the graphical magnification. Use black colour for axes. Differentiate line colours of 200 and 500, in particular for section C. Separate “DPPH” and (%)
Response: Thanks. Figures with high resolution have been added, and a file of higher resolution images are also uploaded separately. The series colors in Fig. 5 have been changed as follows: light blue (for 50) and yellow (for 200) have been changed to bright blue and purple, please see below:
- Line 305: separate "(C)” and “radicals”
Lines 307-310: use “Asterisks indicate significant difference at P < 0.01 (**) and P < 0.05 (*) according to LSD test. Vitamin C (Vc) was assayed as a positive control. Deionized water (0) was used as a negative control.” instead of “According … activity.”
Response: Thanks. They have been revised as suggested.
- Table 1: I prefer the arrangement as a figure.
Why positive (vitamin C) and negative (deionized water) controls are not present?
Line 321: use “Table 1. Percentage of the in vitro scavenging activity of intracellular polysaccharide (IPS) and extracellular polysaccharide (EPS) produced by Lentinula edodes 808 (ACCC 52357)*” instead of “Table 1 … cultures”
Response: Thanks for the comments. As the reviewer pointed out that figure is more intuitively, the table has been replaced by a figure (please see below), and the data of positive control has been added to the figure. The data of the negative control are used in the formula to calculate the data of samples (see section 2.6), and there is no final data of negative controls.
- Lines 322-324: use “* Liquid cultures were performed with 200 mg/L Fe2+ added after 7 days of shaking at 150 rpm and 3 days of static culture and incubated under static conditions at 25 °C for 15, 30 and 45 days. Data are the means of triplicate experiment ± SD. Values in bold are significantly different (P < 0.01) according to the LSD test. nd: not detected” instead of “Polysaccharides … detected”
Response: Thanks. It has been modified.
- Improve the discussion section.
Lines 330-332: improve
Response: Thanks for the suggestions. The discussion section has been improved, and several references have been cited in the revise manuscript.
- Arrange the “Reference” list following the Journal instructions for authors.
Response: Thanks. The reference has been arranged following the Instructions for authors on the website of https://www.mdpi.com/authors/references.

Reviewer 2 Report (Previous Reviewer 2)
I appreciate very much the efforts that the authors have devoted to the improvement of their manuscript. I have no more questions.
Author Response
Dear Reviewer,
Thank you for for reviewing our manuscript.
Best Wishes
Sincerely,
Quanju Xiang, on behalf of all authors
Round 2
Reviewer 1 Report (Previous Reviewer 1)
Some minor adjustments are still needed to make the manuscript fluent and clear.
For the noun referring to an abnormal tissue growth, tumor is the preferred spelling in American English. Tumour is the standard spelling outside America.
Lines 228-229, 250-251, 270-271: use “Asterisks indicate significant statistical difference at p < 0.01 (**) and p < 0.05 (*) according to LSD test.” Instead of “Asterisks … test.”
Line 272: use “Table 1. Pearson correlation coefficients between intracellular polysaccharide (IPS) and exopolysaccharide (EPS) content and phosphoglucose isomerase (PGI), phosphoglucomutase (PGM), and UDPG-pyrophosphorylase (UGP) activities”
Figure 5 section C: verify the “O2-” formatting.
Line 316: use “Figure” instead of “Fig.”
Line 318: use “6B” instead of “B”
Line 322: use “6F” instead of “F”
Lines 326-333: formatting adequately.
Kind regards
Author Response
Point-by-point responses to the Reviewers' comments:
1 Some minor adjustments are still needed to make the manuscript fluent and clear.
Response: We are very appreciated for your comments, which will greatly improve the quality of the manuscript. All of them have been modified.
2 For the noun referring to an abnormal tissue growth, tumor is the preferred spelling in American English. Tumour is the standard spelling outside America.
Response: Thanks for pointing out this, and the “tumor” has been replaced by “tumour” throughout the manuscript.
3 Lines 228-229, 250-251, 270-271: use “Asterisks indicate significant statistical difference at p < 0.01 (**) and p < 0.05 (*) according to LSD test.” Instead of “Asterisks … test.”
Response: Thanks. This statement has been revised in all relevant figures.
4 Line 272: use “Table 1. Pearson correlation coefficients between intracellular polysaccharide (IPS) and exopolysaccharide (EPS) content and phosphoglucose isomerase (PGI), phosphoglucomutase (PGM), and UDPG-pyrophosphorylase (UGP) activities”
Response: Thanks, it has been changes as suggested.
5 Figure 5 section C: verify the “O2-” formatting.
Response: Thanks for pointing out this mistake, it has been changed to “O2-” in figure 5 and 6.
6 Line 316: use “Figure” instead of “Fig.”
Line 318: use “6B” instead of “B”
Line 322: use “6F” instead of “F”
Response: Thanks, all of them have been modified.
7 Lines 326-333: formatting adequately.
Response: Thanks, the legend of figure 6 has been revised.

This manuscript is a resubmission of an earlier submission. The following is a list of the peer review reports and author responses from that submission.
Round 1
Reviewer 1 Report
The manuscript bioengineering-1836916-peer-review-v1 “Enhanced effects of iron on mycelial growth, metabolism and in vitro antioxidant activity of polysaccharide of Lentinus edodes” presents interesting data on the production of polysaccharides by Lentinus edodes in liquid fermentation under different Fe2+ additions.
The structure is in part taken up from the paper cited as [17].
Its presentation in the form of a manuscript is exposed in a cumbersome way and requires several adjustments.
As the first comment, the manuscript does not follow the format proposed by the journal. All references must be arranged as indicated by the authors guidelines.
It is not clear if the manuscript reports on a specific polysaccharide present in two different points of fungal hyphae or different polysaccharides or lentinan.
If I correctly read the paper, Fe2+, as FeCl2, was added to the Lentinus edodes liquid synthetic medium (probably, as reported by reference [17]: 35 g glucose, 5 g peptone, 2.5 g yeast extract, 1 g KH2PO4·H2 O, 0.5 g MgSO4·7H2O, 0.05 g vitamin B1, and 1 L distilled water, sterilized at 121 °C for 30 min). Different Fe2+ concentrations and addition times were tested. Fe2+ was assayed at 0, 50, 200, 600 and 1000 mg/L. Fe2+ was added at: i) inoculation time, ii) after 3 days of static culture, iii) after 7 days of shaking at 150 rpm, iv) after 7 days of shaking (150 rpm) and further 10 days of static culture. These cultures were incubated at 28 °C for 50 days as static (?) cultures. Biomass and polysaccharide production were determined at the end of incubation time. Intracellular (IPS), extracellular (EPS) and total (TPS = %IPS × mg biomass + mg/mL EPS × 50 mL) polysaccharides were investigated. These preliminary experiments allowed the optimal Fe2+induction concentration (200 mg/L) and addition time (after 3 days of static culture). New cultures were performed to analyse changes in mycelial biomass and polysaccharide content over 15, 30, 45 and 60 days. Under the optimal concentration (200 mg/L) and induction time (after 3 days of static culture), the high biomass content (0.28 mg) was collected at 60 d of culture. The presence of Fe2+ in the growth medium also enhanced IPS and EPS concentrations. The activity of enzymes involved in polysaccharide synthesis, including phosphoglucomutase (PGM), phosphoglucose isomerase (PGI) and UDPG-pyrophosphorylase (UGP), was also affected by the addition of Fe in the medium. The activities of PGM and UGP showed a strong correlation (P<0.01) with IPS production. PGM, PGI and UGP genes were up-regulated at 45 days of culture in medium with Fe2+. EPS produced in media Fe-added exhibited a better capacity in scavenging ability on -OH radicals at a very low concentration (0.06 mg/mL). L. edodes cropped with iron increased biomass and polysaccharide production. The polysaccharides produced in media amended with different concentrations of Fe reduced their scavenging effects. In comparison with the control cultures without Fe addition, an increment of biological effects was recorded in the scavenging -OH activity of EPS during all the tested times course.
Some suggestions to improve the manuscript.
Verify the correct use of the International System of Units or uniform the symbols. Use ml (lines 24, 139, 233, 252, 311) or mL (lines 75, 97, 98, 100, 102).
Lines 93, 95, 99, 106: What do “rpm/min” mean in these sentences? “rpm” is the acronym of revolutions per minute and is the number of turns in one minute. It is a unit of rotational speed. Use simply “rpm”
I think that the abstract is a very important part of the manuscript, after reading the manuscript I suggest enhancing it.
The “Introduction” section is pertinent, but the presentation needs to be improved.
Lines 66-68: “The results provide a theoretical basis to develop further iron-polysaccharide complex production in L. edodes and other fungal species.” This sentence is pertinent to the conclusion. Delete or re-write.
The “Material and methods” section is confusing and difficult to read. Re-write.
I suggest the subsections:
2.1. Fungal strain, biomass evaluation and polysaccharides estimation
2.2. Assessment of optimal Fe2+ concentration and induction time
2.3. Biomass and polysaccharides production
2.4. Effects of Fe2+ on enzymes associated with polysaccharides synthesis
2.4.a. activity
2.4.b. transcriptional expression
2.5. in vitro antioxidant activities of polysaccharides
2.6. statistical analysis
Line 72: What does “Experimental” mean in these sentences?
Line 75: What does “synthetic medium” mean in these sentences? Insert composition or indications to reference [17].
Lines 79: indicate the culture typology adopted during the 50 days after the Fe supply. Explain if 50 days including the days before Fe supply or are only after Fe addition.
Lines 81-85: the sentences “, and the changes in biomass, polysaccharide content, enzyme activities, transcriptional expression, and in vitro anti-oxidation ability of extracted polysaccharides were determined every 15 days. In addition, the in vitro anti-oxidation abilities of polysaccharides extracted from cultures with different Fe2+ concentrations were determined.” are not pertinent to the subsection.
Lines 87-89: How biomass was “captured on pre-dried and weighed filter paper”?
Line 99: indicate the centrifuge used.
Lines 101: What does “TPS” mean in this sentence?
Line 104: Indicate the quantity of mycelium used and the ratio with the buffer.
Lines 134-162: Which polysaccharides were used? How were the polysaccharides produced?
The “Results” section is confusing and difficult to read. Re-write following the subsections suggested in the “Material and methods” section.
How ANOVA correlate enzyme activity and polysaccharide (which of the three?) content?
Was pH of cultures affected by Fe additions?
Figure 1:
Sections B and E: use histograms instead of lines.
Section B: verify the dimensions.
The presence of “0” in all the sections, but with two different significances (“0” Fe in sections A, B and C and Fe2+ added at the time of inoculation in sections D, E, F) induces confusion. I suggest the acronyms i, ii, iii, and iv in sections D, E, F instead of 0, 1, 2, and 3, respectively.
What does “g” mean in sections A and D? You will consider “g/mL” or “g/50 mL”
What does “%” mean in sections B and E?
I suggest the legend:
Figure 1. Effects of Fe2+ concentrations (A, B and C) and induction times (D, E and F) on biomass, extracellular (EPS), intracellular (IPS) and total (TPS) polysaccharides production by Lentinula edodes strain 808 (ACCC 52357). CK: control without Fe2+; i: Fe2+ added at the time of inoculation; ii: static culture for 3 days before adding Fe2+; iii: shaking for 7 days before adding Fe2+; iv: shaking for 7 days and static culture for 10 days prior adding Fe2+. Data are the means of ??? replicates ±???. Different letters above the histograms indicate statistical differences according to ??????.
Figure 2
What does “g” mean in section A?
Use “Fe2+” instead of “metal”
Insert the values at the time “0” (starting the experiment) and time “3” when Fe was added. Then use 18, 33, 48 and 63 days of culture.
Reference on statistical significance is suitable.
I suggest the legend:
Figure 2. Dynamic of biomass, extracellular (EPS), intracellular (IPS) and total (TPS) polysaccharides production by Lentinula edodes strain 808 (ACCC 52357) during 60 (63) days of ???? cultures. CK: control without Fe2+; Fe: 200 mg/L of Fe2+. Data are the means of ??? replicates ±???.
Figure 3
Use “Fe2+” instead of “Fe2+”
Insert the values at the time “0” (starting the experiment) and time “3” when Fe was added. Then use 18, 33, 48 and 63 days of culture.
Reference on statistical significance is suitable.
I suggest the legend:
Figure 3. Dynamic of phosphoglucose isomerase (A), phosphoglucomutase (B), and UDPG-pyrophosphorylase (C) production by Lentinula edodes 808 (ACCC 52357) during 60 (63) days of???? cultures. CK: control without Fe2+; Fe: 200 mg/L of Fe2+. Data are the means of ??? replicates ±???.
Figure 4
Use “Fe2+” instead of “metal”
Insert the values at the time “0” (starting the experiment) and time “3” when Fe was added. Then use 18, 33, 48 and 63 days of culture.
Reference on statistical significance is suitable.
I suggest the legend:
Figure 4. Transcriptional expression levels of pgi (encoding phosphoglucose isomerase), pgm (encoding phosphoglucomutase), and ugp (encoding UDPG-pyrophosphorylase) by Lentinula edodes strain 808 (ACCC 52357) during 60 (63) days of ???? cultures. CK: control without Fe2+; Fe: 200 mg/L of Fe2+. Data are the means of ??? replicates ±???.
Line 229: How the polysaccharides were treated with iron?
Figures 5 and 6
Insert or highlight the deviation/error standard.
Reference on statistical significance is suitable.
Re-write the legend.
Data reported in the sections 3.5 are not in agreement with data indicated in the graphics.
The discussion section is confusing. Improve it.
Kind regards
Author Response
Reviewer 2
The manuscript bioengineering-1836916-peer-review-v1 “Enhanced effects of iron on mycelial growth, metabolism and in vitro antioxidant activity of polysaccharide of Lentinus edodes” presents interesting data on the production of polysaccharides by Lentinus edodes in liquid fermentation under different Fe2+ additions.
The structure is in part taken up from the paper cited as [17].
Response: Thanks. As the reviewer pointed out that, most of the present works were carried out with the same idea and experimental methods as reported in our previous work. The results showed that Fe2+ had a better promotion effect on the biosynthesis of polysaccharide from Lentinus edodes mycelium than the two reported metal ions. Therefore, we further determined their in vitro antioxidant activities in present work.
Its presentation in the form of a manuscript is exposed in a cumbersome way and requires several adjustments.
Response: Thanks, we had revised the manuscript by the help of a native English speaker and the structure of the manuscript had been adjusted according to the reviewers' comments.
As the first comment, the manuscript does not follow the format proposed by the journal. All references must be arranged as indicated by the authors guidelines.
Response: Thanks for pointing out this. The references had been arranged according to the author instructions.
It is not clear if the manuscript reports on a specific polysaccharide present in two different points of fungal hyphae or different polysaccharides or lentinan.
Response: Thanks. The differences between the polysaccharides extracted from the two media (with or without Fe2+) are still unclear. Whether the polysaccharides contained Fe2+, how much Fe2+ it contained, and whether the two polysaccharides are structurally different, these questions are currently under analysis in our group. Since the two sources (with or without Fe2+) of polysaccharides exhibited different in vitro antioxidant activities, the structure should be different. Only after analyzing the differences in structure can the essence of the polysaccharides be fully defined.
If I correctly read the paper, Fe2+, as FeCl2, was added to the Lentinus edodes liquid synthetic medium (probably, as reported by reference [17]: 35 g glucose, 5 g peptone, 2.5 g yeast extract, 1 g KH2PO4·H2 O, 0.5 g MgSO4·7H2O, 0.05 g vitamin B1, and 1 L distilled water, sterilized at 121 °C for 30 min). Different Fe2+ concentrations and addition times were tested. Fe2+ was assayed at 0, 50, 200, 600 and 1000 mg/L. Fe2+ was added at: i) inoculation time, ii) after 3 days of static culture, iii) after 7 days of shaking at 150 rpm, iv) after 7 days of shaking (150 rpm) and further 10 days of static culture. These cultures were incubated at 28 °C for 50 days as static (?) cultures. Biomass and polysaccharide production were determined at the end of incubation time. Intracellular (IPS), extracellular (EPS) and total (TPS = %IPS × mg biomass + mg/mL EPS × 50 mL) polysaccharides were investigated. These preliminary experiments allowed the optimal Fe2+induction concentration (200 mg/L) and addition time (after 3 days of static culture). New cultures were performed to analyse changes in mycelial biomass and polysaccharide content over 15, 30, 45 and 60 days. Under the optimal concentration (200 mg/L) and induction time (after 3 days of static culture), the high biomass content (0.28 mg) was collected at 60 d of culture. The presence of Fe2+ in the growth medium also enhanced IPS and EPS concentrations. The activity of enzymes involved in polysaccharide synthesis, including phosphoglucomutase (PGM), phosphoglucose isomerase (PGI) and UDPG-pyrophosphorylase (UGP), was also affected by the addition of Fe in the medium. The activities of PGM and UGP showed a strong correlation (P<0.01) with IPS production. PGM, PGI and UGP genes were up-regulated at 45 days of culture in medium with Fe2+. EPS produced in media Fe-added exhibited a better capacity in scavenging ability on -OH radicals at a very low concentration (0.06 mg/mL). L. edodes cropped with iron increased biomass and polysaccharide production. The polysaccharides produced in media amended with different concentrations of Fe reduced their scavenging effects. In comparison with the control cultures without Fe addition, an increment of biological effects was recorded in the scavenging -OH activity of EPS during all the tested times course.
Some suggestions to improve the manuscript.
Verify the correct use of the International System of Units or uniform the symbols. Use ml (lines 24, 139, 233, 252, 311) or mL (lines 75, 97, 98, 100, 102).
Response: Thanks, we have corrected and uniform the symbol “mL” in the manuscript and checked throughout the full text and figures.
Lines 93, 95, 99, 106: What do “rpm/min” mean in these sentences? “rpm” is the acronym of revolutions per minute and is the number of turns in one minute. It is a unit of rotational speed. Use simply “rpm”
Response: Thanks very much for pointing out this mistake, all of them have been corrected in the revised manuscript.
I think that the abstract is a very important part of the manuscript, after reading the manuscript I suggest enhancing it.
The “Introduction” section is pertinent, but the presentation needs to be improved.
Response: Thanks. The data, statements and language of the Abstract and Introduction had been improved in the revised manuscript.
Lines 66-68: “The results provide a theoretical basis to develop further iron-polysaccharide complex production in L. edodes and other fungal species.” This sentence is pertinent to the conclusion. Delete or re-write.
Response: Thanks. This sentence had been modified as “The results provide a theoretical basis to promote polysaccharide production in L. edodes.” in the revised manuscript.
The “Material and methods” section is confusing and difficult to read. Re-write.
I suggest the subsections:
2.1. Fungal strain, biomass evaluation and polysaccharides estimation
2.2. Assessment of optimal Fe2+ concentration and induction time
2.3. Biomass and polysaccharides production
2.4. Effects of Fe2+ on enzymes associated with polysaccharides synthesis
2.4.a. activity
2.4.b. transcriptional expression
2.5. in vitro antioxidant activities of polysaccharides
2.6. statistical analysis
Response: Thanks for your suggestions. We have modified this section as suggested as follows:
2.1. Fungal strain and culture conditions
2.2. Assessment of optimal Fe2+ concentration and induction time
2.3. Biomass and polysaccharides production
2.4. Effects of Fe2+ on enzymes associated with polysaccharides synthesis
2.4.a. Enzyme activity assays
2.4.b. Transcriptional expression analysis
2.5. In vitro antioxidant activities of polysaccharidess
2.6. Statistical analysis
Line 72: What does “Experimental” mean in these sentences?
Response: Thanks. The subsection title had been modified as “Fungal strain and….” in the revised manuscript.
Line 75: What does “synthetic medium” mean in these sentences? Insert composition or indications to reference [17].
Response: Thanks. The composition of the medium had been inserted in the revised manuscript.
Lines 79: indicate the culture typology adopted during the 50 days after the Fe supply. Explain if 50 days including the days before Fe supply or are only after Fe addition.
Response: Thanks. We are sorry for this unclear description. The culture time had been modified as “The medium were incubated at 25 °C for 50 days in total.” in the revised manuscript
Lines 81-85: the sentences “, and the changes in biomass, polysaccharide content, enzyme activities, transcriptional expression, and in vitro anti-oxidation ability of extracted polysaccharides were determined every 15 days. In addition, the in vitro anti-oxidation abilities of polysaccharides extracted from cultures with different Fe2+ concentrations were determined.” are not pertinent to the subsection.
Response: Thanks. These statements had been transferred to the subsection “2.4. Effects of Fe 2+ on enzymes associated with polysaccharides synthesis”.
Lines 87-89: How biomass was “captured on pre-dried and weighed filter paper”?
Response: Thanks. This sentence had been modified as “The fermentation broth was filtered through a 100-mesh screen, washing by distilled water three times, capturing on a pre-dried and weighed filter paper, and drying at 60 °C until constant and weighed.” in the revised manuscript.
Line 99: indicate the centrifuge used.
Response: Thanks. The brand and model of centrifuge had been added as “Eppendorf 5804R” in the revised manuscript.
Lines 101: What does “TPS” mean in this sentence?
Response: Thanks. “TPS” means total polysaccharide, and the full name had been added in the first place in the revised manuscript.
Line 104: Indicate the quantity of mycelium used and the ratio with the buffer.
Response: Thanks for your suggestion. The quantity of mycelium and buffer had been specified as “Fresh mycelium (0.1g) was washed three times with phosphate buffer, ground into powder in liquid nitrogen, and dissolved into 1 mL phosphate buffer (20 mM, pH 6.5)…….” in the revised manuscript.
Lines 134-162: Which polysaccharides were used? How were the polysaccharides produced?
Response: We are sorry for this confusion. The polysaccharides used for the in vitro antioxidant activities tests had been described in the corresponding part in “2.5. In vitro antioxidant activities of polysaccharides” and “3.4. Effect of Fe2+ on the in vitro antioxidant activity of polysaccharide” in the revised manuscript.
The “Results” section is confusing and difficult to read. Re-write following the subsections suggested in the “Material and methods” section.
How ANOVA correlate enzyme activity and polysaccharide (which of the three?) content?
Response: Thanks. The arrangements of the Results section had been modified corresponding to the “Material and methods” section. And the statement “ANOVA correlate enzyme activity and polysaccharide (which of the three?) content….” is incorrect, and it had been corrected as “Pearson correlation coefficient analysis…..” in the revised manuscript.
Was pH of cultures affected by Fe additions?
Response: Thanks. We did not test whether the addition of metals affects pH. In general, the addition of metals affects the secondary metabolism of microorganisms and may have an effect on pH.
Figure 1:
Sections B and E: use histograms instead of lines.
Section B: verify the dimensions.
Response: Thanks. The two pictures had been modified to histograms, and all the dimensions had been verified.
The presence of “0” in all the sections, but with two different significances (“0” Fe in sections A, B and C and Fe2+ added at the time of inoculation in sections D, E, F) induces confusion. I suggest the acronyms i, ii, iii, and iv in sections D, E, F instead of 0, 1, 2, and 3, respectively.
Response: Thanks for your valuable suggestions. They induction time points had been modified as “I, II, III, and IV” in all the figures and texts.
What does “g” mean in sections A and D? You will consider “g/mL” or “g/50 mL”
Response: Thanks. It had been modified as “g/50 mL” in all corresponding pictures.
What does “%” mean in sections B and E?
Response: Thanks. The “%” indicated the mass percentage, and they had been modified as “wt %”
I suggest the legend:
Figure 1. Effects of Fe2+ concentrations (A, B and C) and induction times (D, E and F) on biomass, extracellular (EPS), intracellular (IPS) and total (TPS) polysaccharides production by Lentinula edodes strain 808 (ACCC 52357). CK: control without Fe2+; i: Fe2+ added at the time of inoculation; ii: static culture for 3 days before adding Fe2+; iii: shaking for 7 days before adding Fe2+; iv: shaking for 7 days and static culture for 10 days prior adding Fe2+. Data are the means of ?? replicates ±???. Different letters above the histograms indicate statistical differences according to ???.
Response: Thanks. The legend has been modified as follows “Effects of Fe2+ concentrations (A, B and C) and induction times (D, E and F) on biomass, extracellular (EPS), intracellular (IPS) and total (TPS) polysaccharides production by Lentinula edodes strain 808 (ACCC 52357). CK: control without Fe2+; I: Fe2+ added at the time of inoculation; II: static culture for 3 days before adding Fe2+; III: shaking for 7 days before adding Fe2+; IV: shaking for 7 days and static culture for 10 days prior adding Fe2+. Data are the means of three replicates. Different letters above the histograms indicate statistical differences (p < 0.05) according to one way analysis.”
Figure 2
What does “g” mean in section A?
Use “Fe2+” instead of “metal”
Response: Thanks. The “g” means the mycelium weight (biomass) in the fermentation flask, and it had been modified as “g/50mL”. The “metal” has been replaced by “Fe2+”in all figures in the revised manuscript.
Insert the values at the time “0” (starting the experiment) and time “3” when Fe was added. Then use 18, 33, 48 and 63 days of culture.
Reference on statistical significance is suitable.
I suggest the legend:
Figure 2. Dynamic of biomass, extracellular (EPS), intracellular (IPS) and total (TPS) polysaccharides production by Lentinula edodes strain 808 (ACCC 52357) during 60 (63) days of ???? cultures. CK: control without Fe2+; Fe: 200 mg/L of Fe2+. Data are the means of ??? replicates ±???.
Figure 3
Use “Fe2+” instead of “Fe2+”
Insert the values at the time “0” (starting the experiment) and time “3” when Fe was added. Then use 18, 33, 48 and 63 days of culture.
Reference on statistical significance is suitable.
I suggest the legend:
Figure 3. Dynamic of phosphoglucose isomerase (A), phosphoglucomutase (B), and UDPG-pyrophosphorylase (C) production by Lentinula edodes 808 (ACCC 52357) during 60 (63) days of???? cultures. CK: control without Fe2+; Fe: 200 mg/L of Fe2+. Data are the means of ??? replicates ±???.
Figure 4
Use “Fe2+” instead of “metal”
Insert the values at the time “0” (starting the experiment) and time “3” when Fe was added. Then use 18, 33, 48 and 63 days of culture.
Reference on statistical significance is suitable.
I suggest the legend:
Figure 4. Transcriptional expression levels of pgi (encoding phosphoglucose isomerase), pgm(encoding phosphoglucomutase), and ugp (encoding UDPG-pyrophosphorylase) by Lentinula edodes strain 808 (ACCC 52357) during 60 (63) days of ???? cultures. CK: control without Fe2+; Fe: 200 mg/L of Fe2+. Data are the means of ??? replicates ±???.
Response: Thanks. Based on the optimal induction concentration and time, the Fe2+ was added at time point “IV (time point “3”in the old version)”, which means the Fe2+ was added 10 days after inoculation. Determination of the biomass and polysaccharides production were conducted after the addition of Fe2+, and no data is available for time “0” and “3”. We would like to retain the abscissa as culture time after the addition of Fe2+. And the corresponding statement had been added in the legend of these figures. The statistical analysis had been added in the figures and all the legends had been improved as suggested by the reviewer.
Line 229: How the polysaccharides were treated with iron?
Response: Thank. We are sorry for the incorrect statement. It is not the polysaccharides treated with iron, it had been modified as “polysaccharides extracted from Fe2+ treated samples” throughout the text and figures.
Figures 5 and 6
Insert or highlight the deviation/error standard.
Reference on statistical significance is suitable.
Re-write the legend.
Data reported in the sections 3.5 are not in agreement with data indicated in the graphics.
Response: Thanks for your comments. The deviation/error bars had been added on the graphs. The differences in antioxidant activity of the control and different concentration of Fe2+ treated samples were compared, showing that most of them were significantly different. There are many series in the line chart, and many data points overlap each other, it is hard to label all of them, so we put the relevant statistical analysis into the text, such as “The EPS from all Fe2+ treated samples exhibited a better and significantly different (P<0.05) capacity in the scavenging effects on -OH radicals (Fig.5B)….”, “The EPS without Fe2+ treatments showed higher and significantly different (P<0.01) scavenging effects on O2- radicals …..”
The discussion section is confusing. Improve it.
Response: Thanks. Based on the research results, the Discussion section had been revised and improved.

Reviewer 2 Report
The authors characterized the enhanced growth, metabolism, enzyme activities, and bioactive activities of polysaccharide in Lentinus edodes by Fe2+. The authors also identified the genes responsible for these significant changes with the treatment of additional Fe2+. I believe that the authors provided sufficient background, explained well the established methodology, presented the results using appropriate figures, and concluded appropriately based on available data. I have no major technical concerns but would like to suggest that the authors proofread the entire manuscript to correct the grammatical and editorial errors throughout the entire manuscript.
Author Response
The authors characterized the enhanced growth, metabolism, enzyme activities, and bioactive activities of polysaccharide in Lentinus edodes by Fe2+. The authors also identified the genes responsible for these significant changes with the treatment of additional Fe2+. I believe that the authors provided sufficient background, explained well the established methodology, presented the results using appropriate figures, and concluded appropriately based on available data. I have no major technical concerns but would like to suggest that the authors proofread the entire manuscript to correct the grammatical and editorial errors throughout the entire manuscript.
Response: Thanks for reviewing our manuscript, and we had revised the manuscript with the help of native English speaker, and formatting modifications were also conducted according to the author instruction.

Round 2
Reviewer 1 Report
The revised version of the manuscript bioengineering-1836916-resubmitted was improved but is still very confusing in some parts.
Remain not clear if the manuscript reports on a specific polysaccharide or several polysaccharides: extracellular (EPS), intracellular (IPS) and total (TPS) polysaccharides.
In some parts of the manuscript, Lentinus edodes produce polysaccharide; in other polysaccharides. Please verify and amend.
Abstract
Enhance abstract content.
The cultivation times are confused: shaking for 7 days and static for 10 days in total, or 60 days or 50 days in total. Associate the correct time to the suitable experiment.
Line 12: use “anti-viral properties” instead of “anti-virus”.
Line 12: What does “natural” mean in this sentence?
Line 13: What does “metabolism” mean in this sentence?
Introduction
Generally, the introduction should briefly place the study in a broad context and highlight why it is important. In this context insert information on the effects of sodium and calcium on polysaccharide production of Lentinus edodes (reference 17).
Lines 65-66: rewrite the sentence “To our knowledge, the effects of iron on polysaccharides biosynthesis and biological activities have not been studied to date.” Iron (III) was evaluated with Auricularia auricula (reference 15) and Flammulina velutipes (reference 16). The affirmation is true for Lentinus edodes.
Materials and methods
Section 2.1: indicate how the synthetic liquid medium was inoculated.
Line 77: use “medium surface” instead of “medium”
Lines 80-82: the sentence “The effects of Fe2+ on the strain was tested by adding various concentrations of Fe2+ (FeCl2) at different culture time to the synthetic medium and incubated at 25 °C.” is not pertinent with the subsection. Delete.
In Section 2.2. indicate:
a) when Fe2+ was added in the experiments for concentration assessment.
b) haw, the cultures were incubated during the optimal concentration assessment.
c) haw, the cultures were incubated after Fe addition during the induction time assessment.
d) control or controls performed.
Line 86: insert “as FeCl2” between “added” and “to”
Line 87: What does “in total” mean in this sentence?
Line 87: use “The cultures” instead of “The medium”; use “as static cultures” instead of “in total”
Line 90: the condition (IV) is not clear. Use “after 7 days of shaking at 150 rpm and 3 days of static culture (IV)” instead of “after shaking for 7 days and static culture for 10 days in total (IV)”
Line 90: medium is a Latin word. It is a substantive singular; media is the plural.
Lines 93-95: the sentence “For further dynamic changes, Fe2+ was added at the optimal induction concentration (200 mg/L) and time () to the synthetic medium, and incubated at 25 °C for further 60 days. Samples were collected every 15 days.” is not pertinent to the subsection. Delete.
Line 96: use “determination” instead of “production”
Line 98: What does “fermentation broth” mean in this sentence? I suppose liquid media plus growing mycelia.
Line 99: What does “capturing” mean in this sentence?
Lines 98-100: use “The fungal biomass was collected by filtration through a 100-mesh screen. The raw material was washed with distilled water (??? ml) three times, trapped on a pre-dried and pre-weighed filter paper, and dried at 60 °C until constant weight.” Instead of “The fermentation broth … weighed.”
Line 102: use “10 mL” instead of “50 times volume”
Line 102: What does “refluxed” mean in this sentence?
Line 104: What does “reflux” mean in this sentence?
Line 104: Integrate centrifuge information: ??? manufacturing, ??? City, ??? State, ????
Lines 104-108: use “The aqueous phase was centrifuged (Eppendorf centrifuge 5804R, ??? manufacturing, ??? City, ??? State, 3000 rpm, 30 min, 4 °C). The supernatant was added with four volumes of 95% ethanol and kept overnight at 4 °C. After centrifugation (3000 rpm, 30 min, 4 °C), the supernatant was discarded, excess ethanol evaporated, and the precipitate IPS was dissolved in 2 mL water.” Instead of “The water phase … 2 mL water.”
Line 108: “To extract EPS” starts as a new paragraph.
Line 109: What does “fermentation broth” mean in this sentence? I suppose liquid media after mycelia collection, that usually is reported as “culture filtrate”.
Lines 108-110: use “To extract EPS, 30 mL ethanol were added to 10 mL of culture filtrate and incubated at 4 °C for 12 h. EPS were precipitated by centrifugation at 4000 rpm for 10 min. After removing the supernatant, the precipitate was dissolved in 2 mL water.” Instead of “To extract EPS … water.”
Line 111: “Polysaccharide contents” starts as a new paragraph.
Line 111: Write how IPS and EPS were expressed.
Line 112: “Total polysaccharide” starts as a new paragraph.
Line 114: Insert the subsection “2.4. Dynamic changes in biomass and polysaccharide
For further dynamic changes, Fe2+ was added at the optimal induction concentration (200 mg/L) and time (7 days of shaking at 150 rpm and 3 days of static culture) to the synthetic medium and incubated at 25 °C for further 60 days. Samples were collected every 15 days and analysed for mycelial biomass, ITS, EPS and TPS value.”
Line 115: delete “enzymes associated with”
In this way the sequence of material and methods subsections became:
2.1. Fungal strain and culture conditions
2.2. Assessment of optimal Fe2+ concentration and induction time
2.3. Biomass and polysaccharides determination
2.4. Dynamic changes in biomass and polysaccharides
2.5. Effects of Fe2+ on polysaccharides synthesis
2.5.a. Enzyme activity assays
2.5.b. Transcriptional expression analysis
2.5. In vitro antioxidant activities of polysaccharides
2.6. Statistical analysis
Lines 116-120: use “To investigate the effect of Fe2+ on the activities of the enzymes involved in L. edodes polysaccharide biosynthesis and the transcriptional expression levels of encoding genes, cultures were performed with the optimal induction concentration (200 mg/L) and induction time (7 days of shaking at 150 rpm and 3 days of static culture). After iron addition, cultures were incubated at 25 °C for further 60 days. Samples were collected every 15 days.” instead of “In order to … polysaccharides synthesis.”
Line 122: write how the mycelium was collected, the mycelium/buffer ratio and the temperature of centrifugation.
Lines 122-125: use “Fresh mycelium (0.1 g), collected as described in section 2.3, was washed three times with ?? mL of 20 mM phosphate buffer (pH 6.5), powdered in liquid nitrogen, and dissolved into 1 mL of phosphate buffer (20 mM, pH 6.5). After centrifugation (10,000 rpm, for 15 min, ?? °C), the supernatant was used as a crude enzyme solution.” instead of “ Fresh mycelium … enzyme solution.”
Line 153: use “ITS and EPS” instead of “these polysaccharides”
Lines 153-156: use “Cultures were performed as reported in section 2.2.” instead of “The polysaccharides … biological activities.”
Lines 187-189: ANOVA includes different mean separation techniques (multiple comparison procedures) such as LSD, Tukey, Duncan, etc. Which was applied in your experiments?
Explain how the Pearson correlation coefficient was calculated for polysaccharide contents and enzyme activities.
Results
This section reports information related to material and methods and discussion. Usually, the results section supplies a concise and precise description of the experimental results.
Figures must be self-explanatory: each figure must be sufficiently complete, clear and easy to understand without needing any extra explanation.
Re-write.
Line 193: use “ Fig. 1A, 1B, 1C)” instead of “Fig. 1”
Line 200: maybe that “was” is preferable to “is”.
Figure 1: verify and improve the image resolution.
The legend present in the manuscript differs from that reported in the response to the reviewer.
I suggest the legend:
Figure 1. Effects of Fe2+ concentrations (A, B and C) and induction times (D, E and F) on biomass, extracellular (EPS), intracellular (IPS) and total (TPS) polysaccharides production by Lentinula edodes strain 808 (ACCC 52357). CK: control ??? cultures in liquid medium without Fe2+. Fe2+, as FeCl2, was added: I) at the time of inoculation; II) after 3 days of static culture; III) after 7 days of shaking at 150 rpm; IV) after 7 days of shaking at 150 rpm and 3 days of static culture. After Fe addition flasks were grown in static conditions. All cultures were incubated at 25 °C. Data are recorded after 50 days. Values are the means of three replicates ± SD. Different letters above the histograms indicate statistical differences (p < 0.05) according to ??? test.”
Lines 207-211: use “The experiment allowed to set up the optimal induction time showed (Fig. 1D, 1E, 1F)” instead of “Microorganism … showed”
Line 213: using “7 days of shaking at 150 rpm” instead of “shaking for 7 days”
Lines 213-214: use “7 days of shaking at 150 rpm and 3 days of static culture” instead of “shaking for 7 days and static culture for 10 days in total”
Line 216: delete “production”
Lines 217-221: use “Data on dynamic of mycelia and polysaccharides production under optimal Fe2+ concentration (200 mg/L) and induction time (after 7 days of shaking at 150 rpm and 3 days of static culture and further incubation as static cultures) are reported in Figure 2.” instead of “Specific secondary … over time.
Figure 2.
Considering the presence of acronyms on the axis, A, B, C, and D could be deleted.
Line 228: use “Lentinula” instead of “L.”; use “60” instead of “70”; insert “after Fe addition” between “cultures” and “.”
Lines 229-230: use “Fe2+ added after 7 days of shaking at 150 rpm and 3 days of static culture.” Instead of “Fe2+. The … adding Fe2+.”
Lines 230: use “replicates ± SD” instead of “replicates”
Line 231: use “(p < 0.05) according to ??? test.” Instead of “(p < 0.05)”
Line 233: delete “enzymes associated with”
Lines 234-238: Remove.
Line 239: use “ PGI activity” instead of “The activity of phosphoglucose isomerase (PGI)”
Line 241: “The activity” starts as a new paragraph.
Line 242: remove “phosphoglucomutase (” and “)”
Line 244: “The activity” starts as a new paragraph.
Line 245: remove “UDPG-pyrophosphorylase (” and “)”
Line 247: What does “r2” mean in this sentence? Explain under the “Statistical analysis” section
Figure 3:
Line 251: use “Lentinula” instead of “L.”; use “60” instead of “70”; insert “after Fe addition” between “cultures” and “.”
Lines 252-253: use “Fe2+ added after 7 days of shaking at 150 rpm and 3 days of static culture.” Instead of “Fe2+. The time on the abscissa represents the incubation time after adding Fe2+.”
Lines 253: use “replicates ± SD” instead of “replicates”
Line 254: use “(p < 0.05) according to ??? test.” Instead of “(p < 0.05)”
Lines 257-258: Remove “Furthermore, … PCR.”
Line 259: “The relative” starts as a new paragraph.
Line 261: “The expression” starts as a new paragraph.
Line 263: “The expression” starts as a new paragraph.
Figure 4
Lines 268-272: replace with “Transcriptional expression levels of pgi (A), pgm (B), and ugp (C) genes, by Lentinula edodes 808 (ACCC 52357) during 60 days of static cultures after Fe addition. CK: control without Fe addition; Fe: 200 mg/L of Fe2+ added after 7 days of shaking at 150 rpm and 3 days of static culture. Data are the means of three replicates ± SD. The different lowercase letters indicate that the results are significantly different (p < 0.05) according to ??? test.”
Line 275: delete
Line 276-277: use “different concentrations are presented in figure 5.” instead of “five different concentrations (0, 50, 100, 200, and 500 mg/L) of Fe2+ were determined.”
Line 289: What does “Two-way ANOVA tests” mean in this sentence? If correct, explain under the “Statistical analysis” section.
Figures 5:
Improve the graphical differentiation. Use assorted colours.
What does “Vc” mean in this figure?
Figure legend reports six concentrations 0, 50, 100, 500, 1000, Vc), while the line chars contain five different curves.
Is SD present?
Insert statistical significance.
Lines 292-296: replace with “Figure 5. In vitro scavenging effects of intracellular (IPS; A, C, E) and extracellular (EPS; B, D, F) polysaccharides from Lentinula edodes 808 (ACCC 52357) cultures on –OH (A, B), DPPH (C, D) and O2- (E, F) radicals. Polysaccharides were extracted from liquid static cultures under different Fe2+ concentrations (Vc???, 0, 50, 100, 200, and 500 mg/L) added at the inoculation time and incubated under static conditions at 25 °C for 50 days. Data are the means of three replicates ± SD. The different lowercase letters indicate that the results are significantly different (p < 0.05) according to ??? test. Vc = ????”
Line 297: delete
Lines 298-301: use “The in vitro antioxidant activities of polysaccharide produced by L. edodes liquid cultures performed under optimal Fe2+ concentration and induction time are presented in figure 6.” instead of “In order … determined.”
Line 307: What does “Two-way ANOVA tests” mean in this sentence? If correct, explain under the “Statistical analysis” section.
Figures 6:
Improve the graphical differentiation. Use distinct colours.
What does “Vc” mean in this figure?
Delete “(200)” in the figure legend.
Is SD present?
Insert statistical significance.
Lines 310-314: replace with “Figure 6. In vitro scavenging effects of intracellular (IPS; A, C, E) and extracellular (EPS; B, D, F) polysaccharides from Lentinula edodes 808 (ACCC 52357) cultures on –OH (A, B), DPPH (C, D) and O2- (E, F) radicals. Polysaccharides were extracted from liquid static cultures with 200 mg/L Fe2+ added after 7 days of shaking at 150 rpm and 3 days of static culture and incubated under static conditions at 25 °C for 15, 30 and 45 days. Data are the means of three replicates ± SD. The different lowercase letters indicate that the results are significantly different (p < 0.05) according to ??? test. Vc = ????”
Discussion
Usually, in this section, the Authors discuss the results and how they can be interpreted in the perspective of previous studies and the working hypotheses.
This section remains confusing.
Line 316: What does “main secondary metabolite” mean in this sentence?
Line 317: “anti-oxidation” or “anti-oxidant activity”;
Line 317: What does “immunity” mean in this sentence?
Line 317: What does “natural” mean in this sentence?
Line 320: What does “management” mean in this sentence?
Kind regards
Author Response
Dear editor, dear Reviewer, Thank you for your constructive and helpful comments concerning our manuscript. We have revised our manuscript in response to your comments. Please find a detailed point-by-point reply below. Colour coding: Referee comments are in black, our responses are in blue; in the revised version of the manuscript, all changes are marked using the “Track Changes” function MS Word Sincerely, Quanju Xiang, on behalf of all authors ----------------------------------------------- Point-by-point responses to the Reviewers' comments: 1. The revised version of the manuscript bioengineering-1836916-resubmitted was improved but is still very confusing in some parts. Response: Thanks, we have revised our manuscript in response to your comments and checked the full text. 2. Remain not clear if the manuscript reports on a specific polysaccharide or several polysaccharides: extracellular (EPS), intracellular (IPS) and total (TPS) polysaccharides. Response: Thanks. Both the EPS and IPS were extracted and analyzed in this study. Since the total content of IPS related with the biomass, so the content of total polysaccharide was also analyzed when determining the optimal concentration and induction time. 3. In some parts of the manuscript, L. edodes produce polysaccharide; in other polysaccharides. Please verify and amend. Response: Thanks. We have corrected this, and the “polysaccharide” represents one type of polysaccharide (IPS or EPS), and “polysaccharides” means all kinds of polysaccharide (IPS and EPS). Abstract 4. Enhance abstract content. Response: Thanks, the abstract has been improved. 5. The cultivation times are confused: shaking for 7 days and static for 10 days in total, or 60 days or 50 days in total. Associate the correct time to the suitable experiment. Response: Thanks, we are sorry for the confusion, and the time points has been revised as “at the inoculation (I), after 3 days of static culture (II), after 7 days of shake culture (III), or after 7 days of shake and then 3 days of static culture (IV)” in the revised manuscript. 6. Line 12: use “anti-viral properties” instead of “anti-virus”. Response: Thanks, it has been corrected. 7. Line 12: What does “natural” mean in this sentence? Response: Thanks. It means the content in the fruiting body of Lentinus edodes is low. This sentence has been revised as “……, but their content in L. edodes is very low” for more clear. 8. Line 13: What does “metabolism” mean in this sentence? Response: Thanks. The “metabolism” here refers to the production of polysaccharides and enzymes. 9. Introduction Generally, the introduction should briefly place the study in a broad context and highlight why it is important. In this context insert information on the effects of sodium and calcium on polysaccharide production of Lentinus edodes (reference 17). Response: Thanks. The corresponding information has been added in the revised manuscript. 10. Lines 65-66: rewrite the sentence “To our knowledge, the effects of iron on polysaccharides biosynthesis and biological activities have not been studied to date.” Iron (III) was evaluated with Auricularia auricula (reference 15) and Flammulina velutipes (reference 16). The affirmation is true for Lentinus edodes. Response: Thanks, this statement has been revised as “the effects of iron on polysaccharides biosynthesis and biological activities in L. edodes have not been studied to date” 11. Materials and methods Section 2.1: indicate how the synthetic liquid medium was inoculated. Response: Thanks. A statement “…., and mycelial plugs (5 mm diameter) were used as inoculum. Three mycelial plugs were inoculated into….” has been added in the revised manuscript. 12. Line 77: use “medium surface” instead of “medium” Response: Thanks, it has been corrected. 13. Lines 80-82: the sentence “The effects of Fe2+ on the strain was tested by adding various concentrations of Fe2+ (FeCl2) at different culture time to the synthetic medium and incubated at 25 °C.” is not pertinent with the subsection. Delete. Response: Thanks, it has been deleted in the revised manuscript. 14. In Section 2.2. indicate: a) when Fe2+ was added in the experiments for concentration assessment. Response: Thanks, this sentence has been modified as “…. added to the synthetic medium at the start of cultivation” b) haw, the cultures were incubated during the optimal concentration assessment. Response: Thanks. We added a statement “Samples were collect after 50 days of static culture at 25°C” in the revised manuscript. c) haw, the cultures were incubated after Fe addition during the induction time assessment. Response: Thanks, the sentence has been revised as “the cultures were incubated under static conditions for 50 days at 25 °C” in the revised manuscript. d) control or controls performed. Response: Thanks. For the optimal concentration study, the concentration “0” served as the control, while in the optimal induction time study, 15. Line 86: insert “as FeCl2” between “added” and “to” Response: Thanks, it has been added. 16. Line 87: What does “in total” mean in this sentence? Response: Thanks, this statement has been modified as “The cultures were incubated under static conditions for 50 days at 25 °C”. 17. Line 87: use “The cultures” instead of “The medium”; use “as static cultures” instead of “in total” Response: Thanks, the “medium” has been replaced by “cultures”, and “in total” has been deleted. 18. Line 90: the condition (IV) is not clear. Use “after 7 days of shaking at 150 rpm and 3 days of static culture (IV)” instead of “after shaking for 7 days and static culture for 10 days in total (IV)” Response: Thanks for the valuable suggestion, it has been corrected throughout the manuscript. 19. Line 90: medium is a Latin word. It is a substantive singular; media is the plural. Response: Thanks. 20. Lines 93-95: the sentence “For further dynamic changes, Fe2+ was added at the optimal induction concentration (200 mg/L) and time () to the synthetic medium, and incubated at 25 °C for further 60 days. Samples were collected every 15 days.” is not pertinent to the subsection. Delete. Response: Thanks. These sentences have been deleted in this subsection. 21. Line 96: use “determination” instead of “production” Response: Thanks, it has been modified. 22. Line 98: What does “fermentation broth” mean in this sentence? I suppose liquid media plus growing mycelia. Response: Thanks, it means liquid media plus growing mycelia. 23. Line 99: What does “capturing” mean in this sentence? Response: Thanks, the word “capture” here means collect, this paragraph has been modified as suggested by reviewer. 24. Lines 98-100: use “The fungal biomass was collected by filtration through a 100-mesh screen. The raw material was washed with distilled water (??? ml) three times, trapped on a pre-dried and pre-weighed filter paper, and dried at 60 °C until constant weight.” Instead of “The fermentation broth … weighed.” Response: Thanks for the valuable and helpful comments, and these sentences have been modified as suggested. 25. Line 102: use “10 mL” instead of “50 times volume” Response: Thanks, it has been modified. 26. Line 102: What does “refluxed” mean in this sentence? Response: Thanks, since the polysaccharide was extracted by water extraction and alcohol precipitation. For the IPS extraction, water was added to the mycelium powder and heated. In order to prevent the evaporation of water during heating, and better dissolve the polysaccharides in the mycelium powder, a reflux device is used in this process. 27. Line 104: What does “reflux” mean in this sentence? Response: Thanks. A reflux device has been used during the process of IPS extraction 28. Line 104: Integrate centrifuge information: ??? manufacturing, ??? City, ??? State, ???? Response: Thanks, the Information on centrifuge has been supplemented as “Eppendorf centrifuge 5804R, Eppendorf, Hamburg, Germany” 29. Lines 104-108: use “The aqueous phase was centrifuged (Eppendorf centrifuge 5804R, ??? manufacturing, ??? City, ??? State, 3000 rpm, 30 min, 4 °C). The supernatant was added with four volumes of 95% ethanol and kept overnight at 4 °C. After centrifugation (3000 rpm, 30 min, 4 °C), the supernatant was discarded, excess ethanol evaporated, and the precipitate IPS was dissolved in 2 mL water.” Instead of “The water phase … 2 mL water.” Response: Thanks, it has been modified as suggested by the reviewer. 30. Line 108: “To extract EPS” starts as a new paragraph. Response: Thanks, it has been stated as a new paragraph. 31. Line 109: What does “fermentation broth” mean in this sentence? I suppose liquid media after mycelia collection, that usually is reported as “culture filtrate”. Response: Thanks for the valuable suggestion, it has been changed as “culture filtrate”. 32. Lines 108-110: use “To extract EPS, 30 mL ethanol were added to 10 mL of culture filtrate and incubated at 4 °C for 12 h. EPS were precipitated by centrifugation at 4000 rpm for 10 min. After removing the supernatant, the precipitate was dissolved in 2 mL water.” Instead of “To extract EPS … water.” Response: Thanks, it has been changed. 33. Line 111: “Polysaccharide contents” starts as a new paragraph. Response: Thanks. it has been stated as a new paragraph. 34. Line 111: Write how IPS and EPS were expressed. Response: Thanks. A statement “The content of IPS is expressed as a percentage of biomass ( wt %), and EPS is expressed in milligrams of polysaccharides per milliliter (mg/mL).” has been added in the revised manuscript. 35. Line 112: “Total polysaccharide” starts as a new paragraph. Response: Thanks, it has been stated as a new paragraph. 36. Line 114: Insert the subsection “2.4. Dynamic changes in biomass and polysaccharide For further dynamic changes, Fe2+ was added at the optimal induction concentration (200 mg/L) and time (7 days of shaking at 150 rpm and 3 days of static culture) to the synthetic medium and incubated at 25 °C for further 60 days. Samples were collected every 15 days and analysed for mycelial biomass, ITS, EPS and TPS value.” Response: Thanks, the subsection has been added. 37. Line 115: delete “enzymes associated with” In this way the sequence of material and methods subsections became: 2.1. Fungal strain and culture conditions 2.2. Assessment of optimal Fe2+ concentration and induction time 2.3. Biomass and polysaccharides determination 2.4. Dynamic changes in biomass and polysaccharides 2.5. Effects of Fe2+ on polysaccharides synthesis 2.5.a. Enzyme activity assays 2.5.b. Transcriptional expression analysis 2.5. In vitro antioxidant activities of polysaccharides 2.6. Statistical analysis Response: Thanks. This section has been modified. 38. Lines 116-120: use “To investigate the effect of Fe2+ on the activities of the enzymes involved in L. edodes polysaccharide biosynthesis and the transcriptional expression levels of encoding genes, cultures were performed with the optimal induction concentration (200 mg/L) and induction time (7 days of shaking at 150 rpm and 3 days of static culture). After iron addition, cultures were incubated at 25 °C for further 60 days. Samples were collected every 15 days.” instead of “In order to … polysaccharides synthesis.” Response: Thanks. It has been modified. 39. Line 122: write how the mycelium was collected, the mycelium/buffer ratio and the temperature of centrifugation. Lines 122-125: use “Fresh mycelium (0.1 g), collected as described in section 2.3, was washed three times with ?? mL of 20 mM phosphate buffer (pH 6.5), powdered in liquid nitrogen, and dissolved into 1 mL of phosphate buffer (20 mM, pH 6.5). After centrifugation (10,000 rpm, for 15 min, ?? °C), the supernatant was used as a crude enzyme solution.” instead of “ Fresh mycelium … enzyme solution.” Response: Thanks. This sentence has been change as suggested by reviewer. 40. Line 153: use “ITS and EPS” instead of “these polysaccharides” Response: Thanks. It has been modified. 41. Lines 153-156: use “Cultures were performed as reported in section 2.2.” instead of “The polysaccharides … biological activities.” Response: Thanks. Thanks. It has been changed. 42. Lines 187-189: ANOVA includes different mean separation techniques (multiple comparison procedures) such as LSD, Tukey, Duncan, etc. Which was applied in your experiments? Response: Thanks. This sentence has been changed as “analysis of variance (ANOVA, St. Louis, MA, USA) including least significant difference (LSD)” in the revised manuscript. 43. Explain how the Pearson correlation coefficient was calculated for polysaccharide contents and enzyme activities. Response: Thanks. The three biological replicates of enzyme activities at each time point and the corresponding polysaccharide content were analyzed using Pearson's correlation coefficient by the software SPSS. Results This section reports information related to material and methods and discussion. Usually, the results section supplies a concise and precise description of the experimental results. Figures must be self-explanatory: each figure must be sufficiently complete, clear and easy to understand without needing any extra explanation. Re-write. Response: Thanks for the valuable suggestion, this section has been improved. 44. Line 193: use “ Fig. 1A, 1B, 1C)” instead of “Fig. 1” Response: Thanks, it has been revised. 45. Line 200: maybe that “was” is preferable to “is”. Response: Thanks, it has been revised. 46. Figure 1: verify and improve the image resolution. Response: Thanks. Figures with high resolution (vectorgraph) have been uploaded separately. 47. The legend present in the manuscript differs from that reported in the response to the reviewer. I suggest the legend: Figure 1. Effects of Fe2+ concentrations (A, B and C) and induction times (D, E and F) on biomass, extracellular (EPS), intracellular (IPS) and total (TPS) polysaccharides production by Lentinula edodes strain 808 (ACCC 52357). CK: control ??? cultures in liquid medium without Fe2+. Fe2+, as FeCl2, was added: I) at the time of inoculation; II) after 3 days of static culture; III) after 7 days of shaking at 150 rpm; IV) after 7 days of shaking at 150 rpm and 3 days of static culture. After Fe addition flasks were grown in static conditions. All cultures were incubated at 25 °C. Data are recorded after 50 days. Values are the means of three replicates ± SD. Different letters above the histograms indicate statistical differences (p < 0.05) according to ??? test.” Response: Thanks, it has been revised as “Figure 1. Effects of Fe2+ concentrations (A, B and C) and induction times (D, E and F) on biomass, extracellular (EPS), intracellular (IPS) and total (TPS) polysaccharides production by Lentinula edodes strain 808 (ACCC 52357). CK: control, cultures in liquid medium without Fe2+. Fe2+, as FeCl2, was added: I) at the time of inoculation; II) after 3 days of static culture; III) after 7 days of shaking at 150 rpm; IV) after 7 days of shaking at 150 rpm and 3 days of static culture. After Fe addition flasks were grown in static conditions. All cultures were incubated at 25 °C. Data are recorded after 50 days. Values are the means of three replicates ± SD. Different letters above the histograms indicate statistical differences (p < 0.05) according to ANOVA-LSD test.” 48. Lines 207-211: use “The experiment allowed to set up the optimal induction time showed (Fig. 1D, 1E, 1F)” instead of “Microorganism … showed” Line 213: using “7 days of shaking at 150 rpm” instead of “shaking for 7 days” Lines 213-214: use “7 days of shaking at 150 rpm and 3 days of static culture” instead of “shaking for 7 days and static culture for 10 days in total” Line 216: delete “production” Response: Thanks. They have been deleted. 49. Lines 217-221: use “Data on dynamic of mycelia and polysaccharides production under optimal Fe2+ concentration (200 mg/L) and induction time (after 7 days of shaking at 150 rpm and 3 days of static culture and further incubation as static cultures) are reported in Figure 2.” instead of “Specific secondary … over time. Response: Thanks. This sentence has been improved as suggested by the reviewer. 50. Figure 2. Considering the presence of acronyms on the axis, A, B, C, and D could be deleted. Response: Thanks. The detailed information has been stated in the acronyms on the axis, and the “A, B, C, and D” have been deleted. 51. Line 228: use “Lentinula” instead of “L.”; use “60” instead of “70”; insert “after Fe addition” between “cultures” and “.” Lines 229-230: use “Fe2+ added after 7 days of shaking at 150 rpm and 3 days of static culture.” Instead of “Fe2+. The … adding Fe2+.” Lines 230: use “replicates ± SD” instead of “replicates” Line 231: use “(p < 0.05) according to ??? test.” Instead of “(p < 0.05)” Line 233: delete “enzymes associated with” Lines 234-238: Remove. Line 239: use “ PGI activity” instead of “The activity of phosphoglucose isomerase (PGI)” Line 241: “The activity” starts as a new paragraph. Line 242: remove “phosphoglucomutase (” and “)” Line 244: “The activity” starts as a new paragraph. Line 245: remove “UDPG-pyrophosphorylase (” and “)” Response: Thanks. They have been modified. 52. Line 247: What does “r2” mean in this sentence? Explain under the “Statistical analysis” section Response: Thanks. The “r2” represents the Pearson Correlation Coefficient. This value is between -1 and 1. If it is greater than 0, it indicates a positive correlation. If it is below 0, it indicates a negative correlation. If it is equal to 0, there is no correlation between two groups. 53. Figure 3: Line 251: use “Lentinula” instead of “L.”; use “60” instead of “70”; insert “after Fe addition” between “cultures” and “.” Lines 252-253: use “Fe2+ added after 7 days of shaking at 150 rpm and 3 days of static culture.” Instead of “Fe2+. The time on the abscissa represents the incubation time after adding Fe2+.” Lines 253: use “replicates ± SD” instead of “replicates” Line 254: use “(p < 0.05) according to ??? test.” Instead of “(p < 0.05)” Lines 257-258: Remove “Furthermore, … PCR.” Line 259: “The relative” starts as a new paragraph. Line 261: “The expression” starts as a new paragraph. Line 263: “The expression” starts as a new paragraph. Response: Thanks. They have been modified as suggested by the reviewe. 54. Figure 4 Lines 268-272: replace with “Transcriptional expression levels of pgi (A), pgm (B), and ugp (C) genes, by Lentinula edodes 808 (ACCC 52357) during 60 days of static cultures after Fe addition. CK: control without Fe addition; Fe: 200 mg/L of Fe2+ added after 7 days of shaking at 150 rpm and 3 days of static culture. Data are the means of three replicates ± SD. The different lowercase letters indicate that the results are significantly different (p < 0.05) according to ??? test.” Response: Thanks. This sentence has been improved as suggested by the reviewer. 55. Line 275: delete Response: Thanks. It has been deleted. 56. Line 276-277: use “different concentrations are presented in figure 5.” instead of “five different concentrations (0, 50, 100, 200, and 500 mg/L) of Fe2+ were determined.” Response: Thanks. It has been modified. 57. Line 289: What does “Two-way ANOVA tests” mean in this sentence? If correct, explain under the “Statistical analysis” section. Response: Thanks. “Two-way ANOVA tests” was used to analyze the antioxidant activities of polysaccharides relative to polysaccharide concentrations or to polysaccharide samples (from iron treat or untreated culture; from different incubation time). It has been added to the “Statistical analysis” section. 58. Figures 5:Improve the graphical differentiation. Use assorted colours. Response: Thanks, the figure has been improved. 59. What does “Vc” mean in this figure? Response: Thanks. Vitamin C is a substance with strong antioxidant capacity, and is usually used as a positive control when determining antioxidant activity. It has been described in the subsection “2.6. In vitro antioxidant activities of polysaccharides”. 60. Figure legend reports six concentrations 0, 50, 100, 500, 1000, Vc), while the line chars contain five different curves. Response: Thanks for pointing out this. Five concentrations were set, while Vc served as positive control when determining the antioxidant activities of polysaccharide. 61. Is SD present? Insert statistical significance. Response: Thanks. The SD and statistical significances have been added to all the figures. Significant differences compared to controls (no addition of Fe2+) have been marked on the graph. The font colors of the significant difference were labeled as the same color with the series color. 62. Lines 292-296: replace with “Figure 5. In vitro scavenging effects of intracellular (IPS; A, C, E) and extracellular (EPS; B, D, F) polysaccharides from Lentinula edodes 808 (ACCC 52357) cultures on –OH (A, B), DPPH (C, D) and O2- (E, F) radicals. Polysaccharides were extracted from liquid static cultures under different Fe2+ concentrations (Vc???, 0, 50, 100, 200, and 500 mg/L) added at the inoculation time and incubated under static conditions at 25 °C for 50 days. Data are the means of three replicates ± SD. The different lowercase letters indicate that the results are significantly different (p < 0.05) according to ??? test. Vc = ????” Response: Thanks for your valuable comments. The legend of Figure 5 has been modified as “Figure 5. In vitro scavenging effects of intracellular (IPS; A, C, E) and extracellular (EPS; B, D, F) polysaccharides from Lentinula edodes 808 (ACCC 52357) cultures on –OH (A, B), DPPH (C, D) and O2- (E, F) radicals. Polysaccharides were extracted from liquid static cultures under different Fe2+ concentrations (0, 50, 100, 200, and 500 mg/L) added at the inoculation time and incubated under static conditions at 25 °C for 50 days. Data are the means of three replicates ± SD. According to one-way ANOVA-LSD test, significant differences (**, P < 0.01,* P < 0.05) between control and samples added with Fe2+ were marked as the same colour with series in the graph. Vitamin C (Vc) served as positive control for determination of polysaccharide activity.” 63. Line 297: delete Response: Thanks, it has been deleted. 64. Lines 298-301: use “The in vitro antioxidant activities of polysaccharide produced by L. edodesliquid cultures performed under optimal Fe2+ concentration and induction time are presented in figure 6.” instead of “In order … determined.” Response: Thanks. This sentence has been replaced. 65. Line 307: What does “Two-way ANOVA tests” mean in this sentence? If correct, explain under the “Statistical analysis” section. Response: Thanks. The“Two-way ANOVA tests” was used to analyze effects of two factors, polysaccharide concentration and iron concentration, on the antioxidant activities of polysaccharides. 66. Figures 6: Improve the graphical differentiation. Use distinct colours. Response: Thanks, the figure has been improved. 67. What does “Vc” mean in this figure? Response: Thanks. Vitamin C is a substance with strong antioxidant capacity, and is usually used as a positive control when determining antioxidant activity. It has been described in the subsection “2.6. In vitro antioxidant activities of polysaccharides”. 68. Delete “(200)” in the figure legend. Response: Thank, it has been deleted. 69. Is SD present? Insert statistical significance. Thanks. The SD and statistical significances have been added to all the figures. Significant differences compared to samples incubated 15 days after Fe2+addition have been marked on the graph. The font colors of the significant difference were labeled as the same color with the series color. 70. Lines 310-314: replace with “Figure 6. In vitro scavenging effects of intracellular (IPS; A, C, E) and extracellular (EPS; B, D, F) polysaccharides from Lentinula edodes 808 (ACCC 52357) cultures on –OH (A, B), DPPH (C, D) and O2- (E, F) radicals. Polysaccharides were extracted from liquid static cultures with 200 mg/L Fe2+ added after 7 days of shaking at 150 rpm and 3 days of static culture and incubated under static conditions at 25 °C for 15, 30 and 45 days. Data are the means of three replicates ± SD. The different lowercase letters indicate that the results are significantly different (p < 0.05) according to ??? test. Vc = ????” Response: Thanks for your valuable comments. The legend of Figure 6 has been modified as “Figure 6. In vitro scavenging effects of intracellular (IPS; A, C, E) and extracellular (EPS; B, D, F) polysaccharides from Lentinula edodes 808 (ACCC 52357) cultures on –OH (A, B), DPPH (C, D) and O2- (E, F) radicals. Polysaccharides were extracted from liquid static cultures with 200 mg/L Fe2+ added after 7 days of shaking at 150 rpm and 3 days of static culture and incubated under static conditions at 25 °C for 15, 30 and 45 days. Data are the means of three replicates ± SD. According to one-way ANOVA-LSD test, significant differences (**, P < 0.01,* P < 0.05) between 15d and other two samples were marked as the same colour with series in the graph. Vitamin C (Vc) served as positive control for determination of polysaccharide activity.”. 71. Discussion Usually, in this section, the Authors discuss the results and how they can be interpreted in the perspective of previous studies and the working hypotheses. This section remains confusing. Response: Thanks, the “Discussion” section has been improved according to the results and several references have been added. Line 316: What does “main secondary metabolite” mean in this sentence? Response: Thanks, it has been deleted. 72. Line 317: “anti-oxidation” or “anti-oxidant activity”; Response: Thanks, it has been changed as “anti-oxidant activity”. 73. Line 317: What does “immunity” mean in this sentence? Response: Thanks, it means the polysaccharide has the function of improving human immunity. Line 317: What does “natural” mean in this sentence? Response: Thanks, it has been deleted. Line 320: What does “management” mean in this sentence? Response: Thanks, the “cultivation management” means different management methods during cultivation, such as lighting, cultivation materials, et al. It is reported that log-grown Lentinula edodes contained more High-molecular-weight polysaccharides than did substrate-grown Lentinula edodes.